# Gene expression signatures in blood from a West African sepsis cohort define host response phenotypes

Josh G. Chenoweth [1,14] ✉, Carlo Colantuoni[2,14], Deborah A. Striegel[1], Pavol Genzor [1], Joost Brandsma[1], Paul W. Blair[1,3], Subramaniam Krishnan[1], Elizabeth Chiyka[1], Mehran Fazli [1], Rittal Mehta[1], Michael Considine[4], Leslie Cope[4], Audrey C. Knight[2], Anissa Elayadi[1], Anne Fox [5], Ronna Hertzano[6], Andrew G. Letizia[5], Alex Owusu-Ofori[7,8], Isaac Boakye[9], Albert A. Aduboffour[7], Daniel Ansong[10,11], Eno Biney[12], George Oduro [12], Kevin L. Schully[13] & Danielle V. Clark[1]

Our limited understanding of the pathophysiological mechanisms that operate during sepsis is an obstacle to rational treatment and clinical trial design. There is a critical lack of data from low- and middle-income countries where the sepsis burden is increased which inhibits generalized strategies for therapeutic intervention. Here we perform RNA sequencing of whole blood to investigate longitudinal host response to sepsis in a Ghanaian cohort. Data dimensional reduction reveals dynamic gene expression patterns that describe cell type-specific molecular phenotypes including a dysregulated myeloid compartment shared between sepsis and COVID-19. The gene expression signatures reported here define a landscape of host response to sepsis that supports interventions via targeting immunophenotypes to improve outcomes.

Sepsis is a dysregulated immune response with an infection that results in organ injury[1]. The global burden of sepsis is much higher than previously thought with nearly 50 million incidents in 2017 including 11 million deaths[2]. Although our understanding of the sepsis host response has advanced considerably, it has not translated into effective care and management. A major barrier to progress is the broad definition of sepsis syndrome, which encompasses an array of clinical and biological features. There is an urgent need to better understand the heterogeneous dynamic processes that operate during the course of sepsis to identify treatable traits and support the development of new interventions that improve outcomes[3].

Precision-medicine approaches to better define sepsis through 'endotypes', biologically distinct subgroups with a companion diagnostic and targeted treatment, offer promise. The potential to stratify

[1]Austere environments Consortium for Enhanced Sepsis Outcomes (ACESO), The Henry M. Jackson Foundation for the Advancement of Military Medicine, Inc., Bethesda, MD, USA. [2]Department of Neurology, Johns Hopkins School of Medicine, Baltimore, MD, USA. [3]Department of Pathology, Uniformed Services University, Bethesda, MD, USA. [4]Sidney Kimmel Comprehensive Cancer Center, Johns Hopkins University, Baltimore, MD, USA. [5]Naval Medical Research Unit EURAFCENT Ghana detachment, Accra, Ghana. [6]Section on Omics and Translational Science of Hearing, Neurotology Branch, National Institute on Deafness and Other Communication Disorders, National Institutes of Health, Bethesda, MD, USA. [7]Laboratory Services Directorate, Komfo Anokye Teaching Hospital (KATH), Kumasi, Ghana. [8]Department of Clinical Microbiology, Kwame Nkrumah University of Science and Technology (KNUST), Kumasi, Ghana. [9]Research and Development Unit, KATH, Kumasi, Ghana. [10]Child Health Directorate, KATH, Kumasi, Ghana. [11]Department of Child Health, KNUST, Kumasi, Ghana. [12]Accident and Emergency Department, KATH, Kumasi, Ghana. [13]Austere environments Consortium for Enhanced Sepsis Outcomes (ACESO), Biological Defense Research Directorate, Naval Medical Research Command-Frederick, Ft. Detrick, MD, USA. [14]These authors contributed equally: Josh G. Chenoweth, Carlo Colantuoni. ✉e-mail: jchenoweth@aceso-sepsis.org

sepsis subjects by host response is supported by unsupervised analyses of large multi-omics datasets. Whole blood RNA sequencing from a 306-subject Western Europe cohort identified four molecular sepsis endotypes distinguished by 28-day mortality risk[4]. The high mortality Mars1 group displays gene expression phenotypes consistent with immunosuppression. Another study using a 265-subject cohort from the United Kingdom analyzed global gene expression in peripheral blood leukocytes and revealed two distinct clusters denoted sepsis response signature groups 1 and 2 (SRS1 and SRS2)[5]. Similar to the Mars1 endotype, the SRS1 group gene expression profile indicated T-cell exhaustion and endotoxin tolerance. A third study identified three robust subgroups from a combined 14-cohort analysis of host gene expression data coined by the authors as Coagulopathic, Adaptive, and Inflammopathic[6]. These studies and others extend the pathology of sepsis beyond the cytokine storm and specifically highlight the opportunity to target immunophenotypes to improve outcomes[7,8].

To catalyze clinical trials it will be important to generalize sepsis sub-phenotypes to diverse populations[9]. Notably, the highest burden for sepsis has been described for low, and middle-income countries (LMICs) in sub-Saharan Africa and Asia. However, to date, most sepsis clinical research has not focused on LMICs or populations with diverse bacterial, viral, and parasitic infections that differ from Western countries. To address this gap, we characterized the longitudinal host-response to sepsis in a Ghanaian cohort using RNA sequencing of whole blood. Using dimension reduction and latent space exploration techniques with public RNA sequencing data, we define transcriptional phenotypes in a West African cohort that are shared across diverse cohorts and infection contexts. These gene expression phenotypes map a dynamic cellular landscape of host response to sepsis through time. Our results support interventions that target treatable traits of immunophenotypes to promote improved sepsis outcomes.

## Results

### Dynamic gene expression of the cellular host-response to sepsis

Total RNA from peripheral blood was sequenced from 120 subjects in an observational study of sepsis at Komfo Anokye Teaching Hospital (KATH) in Kumasi, Ghana to define host processes that are active during progression and recovery from sepsis (Fig. 1a)[10]. The study included longitudinal blood collection at enrollment (0 h), 6 hours (h), 24 h, 48 h, 72 h, 28 days (d), 6 months (m) and 12 m, followed by host RNA sequencing (Fig. 1b, c). Specimens were not available for all subjects at all points due to death or loss to follow-up resulting in a total of 531 samples sequenced. Total RNA from 42 healthy Ghanaian donors was also sequenced. Of the 120 subjects analyzed by RNA sequencing, 63 subjects survived to day 28, 54 died by day 28 and 3 had unknown mortality data (Fig. 1d and Supplementary Data 1). Following pre-processing and quality control, Principal Component Analysis (PCA) was applied to explore longitudinal host response and identify expression changes associated with 28-day mortality. Principal component 1 (PC1) segregates healthy donors from those who died by day 28 (Fig. 1e). Transcriptomes from subjects that survive beyond 28 days occupy the full range of PC1 values. The average trajectory of the survivors is a net movement toward healthy donors through time. In contrast, host expression from non-survivors shows no net movement across PC1 through time. Visualization of PC1 across time and survival shows a systematic change in the transcriptome of survivors where later time points (28 d, 6 m, 12 m) co-localize with the healthy donors (Fig. 1f). To assess the generality of the dynamic PC1 pattern, we used available public datasets to quantify similarities with our dataset. A feature mapping projection between two datasets can identify and characterize relationships between datasets without the complications of normalization or sample alignment[11,12]. We downloaded public leukocyte gene expression data from healthy donors and subjects that progress to septic shock (Supplementary Fig. 1a)[13]. Projection of these expression data onto PC1 of the Ghana cohort recapitulates the segregation of healthy controls from septic subjects (Supplementary Fig. 1b).

Next, we assessed the relationship between host gene expression in our West African cohort and previously reported sepsis endotypes. To compare host gene expression in our Ghanaian cohort to the sepsis response signatures SRS1 and SRS2, we used the published machine learning tool SepstratifieR[5,14]. This analysis revealed that our cohort was most aligned with the high-mortality SRS1 group and the probability of being in SRS1 decreases over time in subjects that survive (Fig. 1g, h and Supplementary Fig. 1e). Inspection of genes that define the MARS and the Coagulopathic, Inflammopathic and Adaptive sepsis subgroups also shows these patterns resolve through time in survivors (Supplementary Fig. 1c, d)[4,6]. Innate and adaptive immune dysregulation are common phenotypes across these sepsis subgroups. To explore cellular dynamics in the Ghana cohort the CIBERSORT[15] tool was used to deconvolve bulk gene expression patterns into cell-type specific signatures. We used the LM22 cell-type signatures from Newman et al.[15], as well as those reported by Vallania et al.[16], that use training data from a broad array of healthy and sick patients. Unsupervised clustering identifies sample groups characterized by time and outcome (Supplementary Fig. 2a, b). Notably, healthy donors and survivors 28 days beyond enrollment are distinguished from acute survivors and non-survivors. CIBERSORT-derived proportional cell estimates from both classifiers show that neutrophils and CD4+\CD8+ T cells are positively and negatively correlated to PC1, respectively, consistent with neutrophilia and lymphopenia reported previously for sepsis subjects (Supplementary Fig. 2c)[17,18]. Taken together, these results suggest that the major variation in our Ghana dataset describes a sepsis-associated transcriptional signature with cellular components that resolves in survivors through time.

### Transcriptional features in peripheral blood define prognostic and dynamic phenotypes in sepsis

Given our observations, we sought to further reduce dimensions in our data to identify biologically relevant phenotypes. PCA is limited by the constraint that components be orthogonal, and variability is maximized for the earliest components. This often leads to conflation where multiple biological effects can be contained in a single PC. Other tools are needed for the analysis of dynamic, high-dimensional data in heterogeneous illnesses such as sepsis to gain biological insight. Coordinated Gene Activity in Pattern Sets (CoGAPS) is a sparse Bayesian non-negative matrix factorization (NMF) method that decomposes expression data into component patterns[19–21]. CoGAPS can identify gene expression modules that are specific for subgroups of patients, diseases, transient processes, or other clinically relevant features in a complex dataset that PCA may combine into one component[22]. CoGAPS was applied to the data from the Ghana cohort along with the healthy donors to generate 30 patterns that in combination describe the full variation in the data. Inspection of the results by the 28-day mortality outcome reveals multiple dynamic patterns similar to PC1 that resolve through time (Fig. 2a and Supplementary Fig. 3). CoGAPS also identified patterns that distinguish survivors and non-survivors at enrollment (Fig. 2a, g). We used logistic regression adjusted for age and gender to ask if the CoGAPS patterns predict 28-day mortality. This analysis revealed qualitatively distinct pattern types distinguished by those that are prognostic at enrollment versus dynamic patterns that predict mortality only at later time points (Fig. 2b, h). Some patterns offered little or no predictive power across the cohort and were observed only in a small number of samples or subjects (Fig. 2b and Supplementary Fig. 3). Taken together these results suggest that the features of the biology captured by individual CoGAPS patterns could, in combination, define the host response to sepsis. To test this idea, we used a Random Forest model to determine which CoGAPS patterns are most important for the 28-day mortality

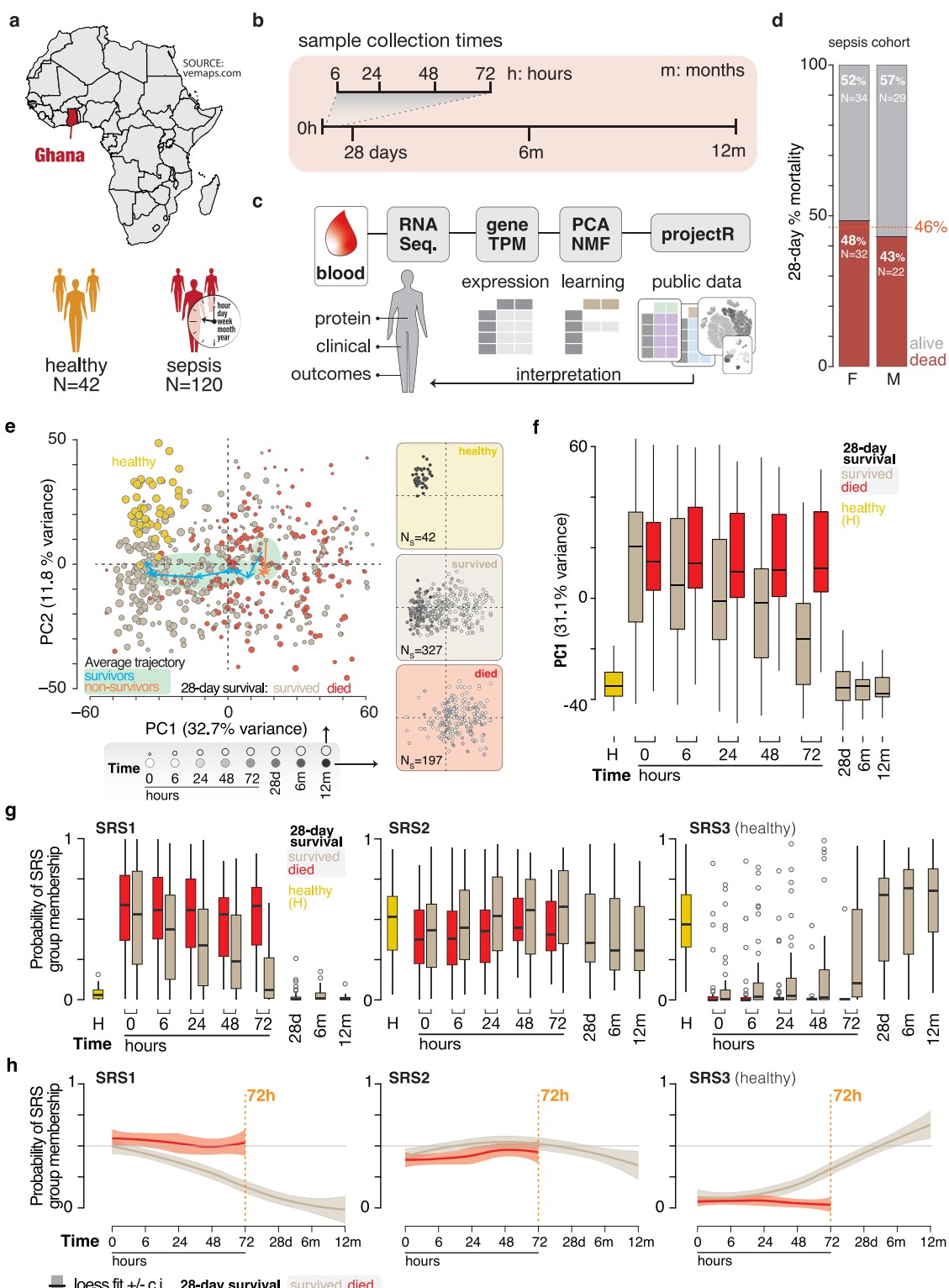

**Fig. 1 | Host gene expression describes recovery in sepsis subjects that survive.**
**a** Description of the study cohort geographic location and size (*N* = subjects). **b** The longitudinal collection time points from subjects starting at enrolment (0 h).
**c** Analytical pipeline used in this study. **d** Breakdown of the sepsis cohort subjects by 28-day mortality and sex. The red line and number show the combined percent 28-day mortality for both sexes. **e** Principal component analysis of gene expression in sepsis subjects and healthy donors. Left plot: the size of the points corresponds to collection time (d: days, m: months). Healthy (gold), surviving (tan), and subjects that died by 28 days (red). Lines with arrows and green shading show the average PCA signal for 28-day survivors (blue), and those who died by 28 days (orange). Right plots: the same three groups of subjects where fill intensity corresponds to collection time points. **f** The

PC1 values of healthy (gold), 28-day surviving (tan), and subjects who died by 28 days (red) at each collection time point. **g** Subjects were analyzed with *SepstratifieR* machine learning algorithm to determine their membership in one of three sepsis response groups (SRS) (Cano-Gamez et al., 2022). **h** Regression analysis (locally weighted least squares [loess] fit line ± confidence intervals [c.i.] as shaded area) with the same groups as in (**g**) depicting the change in SRS group membership over time for 28-day survivors (gray) and those who died by 28 days (red). Exact sample numbers (*n*) for figure panels are described in Supplementary Data 2. The boxplots in (**f**, **g**) describe the median (middle horizontal line), 1st and 3rd quartiles (bottom and top of box, respectively), and data minimum and maximum (vertical whiskers).

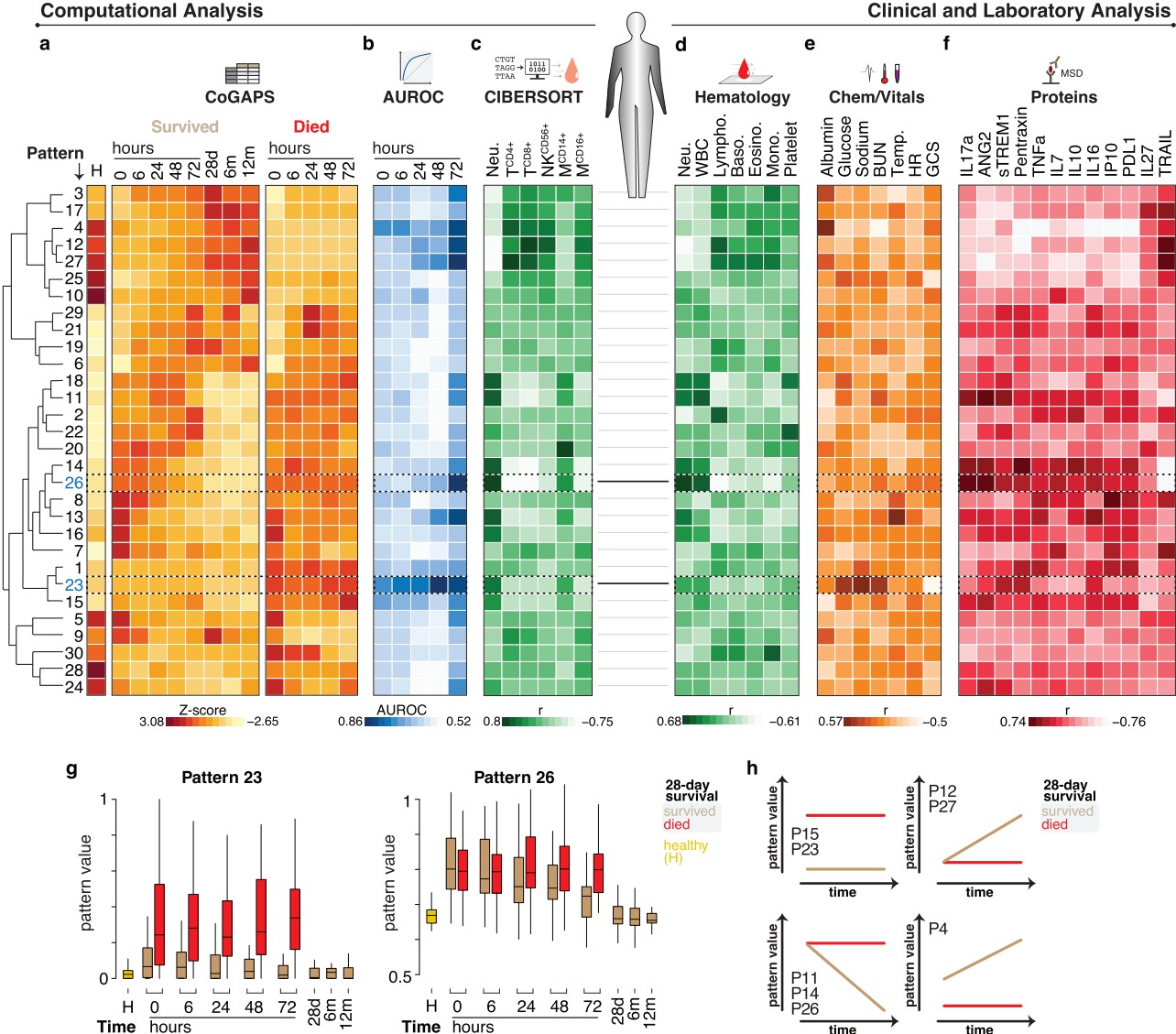

**Fig. 2 | Identification of dynamic and prognostic transcriptional patterns in sepsis and their correlation to laboratory and clinical parameters. a** The study participant gene expression was analyzed using CoGAPS and the resulting thirty (30) patterns were plotted by their z-scores. Shown groups include healthy donors (H), 28-day survivors (tan), and those who died by 28 days (red) over time. The patterns were organized by unsupervised clustering depicted by the dendrogram on the left, and values for each collection time point (0, 6, 24, 48, 72 h, 28 days, 6, and 12 months) are shown on top. **b** Heatmap of AUROC values from logistic regression of CoGAPS patterns fitting to 28-day mortality for the first five time points. **c** Spearman correlation of CIBERSORT ranks to all time points CoGAPS patterns. **d** Hematology results correlated to all available sample time point values (0, 6, and 24 h). **e** Spearman correlation of clinical chemistries and vitals to matched time point CoGAPS values. Chemistries—Albumin, Glucose, Sodium, Blood Urea

Nitrogen (BUN)—were correlated to all available sample time point values (0, 6, and 24 h). Vitals—Temperature (Temp.), heart rate (HR), and Glasgow coma scale (GCS) were correlated to all available sample time point values (0, 6, 24, and 72 h). **f** Spearman correlation of peripheral blood protein measurements to matched CoGAPS pattern values to a matched 6 h time point. **g** Boxplot of selected CoGAPS pattern values—pattern 23 and 26 (dashed boxes in **a**–**e**)—that distinguish survivors from non-survivors at enrollment or during recovery, respectively. **h** Model showing four dynamic and prognostic longitudinal pattern classes that distinguish survivors (tan) from non-survivors (red). Exact sample numbers (*n*) for figure panels are described in Supplementary Data 2. The boxplot in (**g**) describes the median (middle horizontal line), 1st and 3rd quartiles (bottom and top of box respectively), and data minimum and maximum (vertical whiskers).

outcome. This modeling method showed comparatively better performance in predicting 28-day mortality (Supplementary Fig. 4d). Top features include both prognostic (P23, P4, P15) and dynamic patterns (P27, P12, P14) (Supplementary Fig. 4a). Latent space exploration using the public gene expression data from subjects that progress to septic shock generated by Cazalis et al., confirms that these top CoGAPS patterns readily distinguish cases and controls[13] (Supplementary Fig. 3). Notably, pattern 23 more specifically describes septic shock (Supplementary Fig. 4).

To define the biological significance of the CoGAPS patterns we compared individual sample pattern values to clinical and hematology

features. Prognostic patterns 23 and 4 that predict mortality at enrollment correlate most strongly with clinical chemistries and functional assessments including albumin, Blood Urea Nitrogen (BUN), sodium, glucose, and Glasgow Coma Scale (GCS) (Fig. 2e). Comparisons of patterns with CIBERSORT deconvolution (Fig. 2c) and hematology (Fig. 2d), cell proportions show that dynamic patterns 14 and 26 describe neutrophil numbers (Fig. 2d). In contrast, dynamic patterns 27 and 12 monitor CD4+ and CD8 + T-cells\CD56 + NK-cells, respectively. We also looked at circulating factors measured in peripheral blood samples from these subjects (Fig. 2f). Neutrophil-associated patterns 14 and 26 correlate positively with IL-17a and ANG2 and

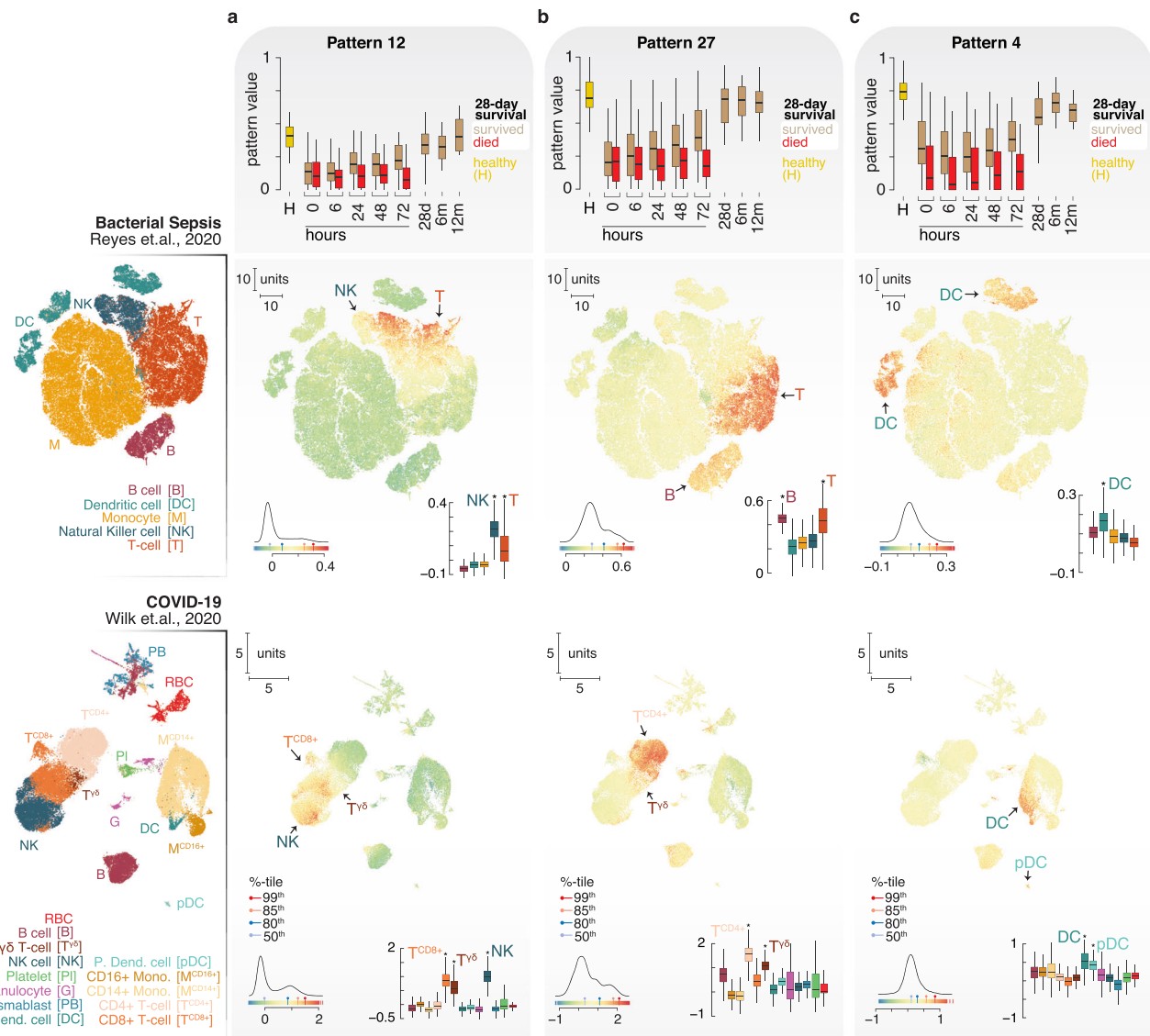

**Fig. 3 | Cellular phenotypes in sepsis subjects.** Projection of published dimensionally reduced uniform manifold approximation and projection (UMAP) single-cell RNA-Seq data describing immune cells in bacterial sepsis (Reyes et al., 2020, PMID: 32066974) and COVID-19 (Wilk et al., 2020, PMID: 32514174) into **a** pattern 12, **b** pattern 27, and **c** pattern 4. The boxplots show pattern value dynamics during the longitudinal course of the study comparing healthy donors (gold), 28-day survivors (tan), and those who died by 28 days (red). The projections are colored by the magnitude of the CoGAPS patterns in individual cells. The color gradient bar shows the range of projected values, percentile pins (%-tile) show selected percentile cutoffs, and the histogram depicts the distribution of all pattern values. Small boxplots at each UMAP projection represent CoGAPS pattern values grouped by cell types. The stars (*) correspond to the significance of cell type enrichment using permutations of group means ($p$-value < 1e−5, see "Methods" for more details). Exact sample numbers ($n$) for figure panels are described in Supplementary Data 2. All boxplots describe the median (middle horizontal line), 1st and 3rd quartiles (bottom and top of box, respectively), and data minimum and maximum (vertical whiskers).

negatively with TRAIL, consistent with neutrophil recruitment in bacterial infections[23]. These results indicate that CoGAPS gene expression patterns are linked to physiological and cellular components of the immune response in sepsis.

## Shared myeloid phenotypes in sepsis and COVID-19

To further define the cellular phenotypes in our cohort, we compared the Ghanaian cohort CoGAPS patterns to single-cell (sc) RNA sequencing datasets from bacterial sepsis[24] and SARS-CoV-2, a major cause of viral sepsis[25,26]. We initially focused on patterns 12, 27, and 4 which are elevated in healthy controls (Fig. 3a–c). Projection of these public sepsis and COVID-19 single-cell datasets onto the CoGAPS patterns supports a lymphocyte phenotype most specific to NK and effector CD8 + T-cells for pattern 12, and CD4 + T-cells and B-cells for pattern 27

(Fig. 3a, b). Pattern 4 best captures a dendritic cell phenotype in both public single-cell datasets (Fig. 3c). Because pattern 4 is both prognostic and dynamic we inspected the single-cell datasets for severity-related phenotypes. Most notably pattern 4 was highly enriched in myeloid cells of healthy individuals in the COVID-19 dataset and was depleted in severely ill subjects (Supplementary Fig. 5a–c). The myeloid compartment has recently been shown to be highly dysregulated in both sepsis and COVID-19 with evidence of immunosuppressed monocytes and dysfunctional neutrophils[27–29]. We looked more closely at myeloid populations in CoGAPS patterns selected by the Random Forest Model that are elevated in sepsis. Pattern 11 is enriched in the recently reported MS3 population that describes classical CD16+ high monocytes while pattern 14 identifies the sepsis and COVID-19-associated CD14+ monocyte state 1 (MS1)[24,27] (Supplementary

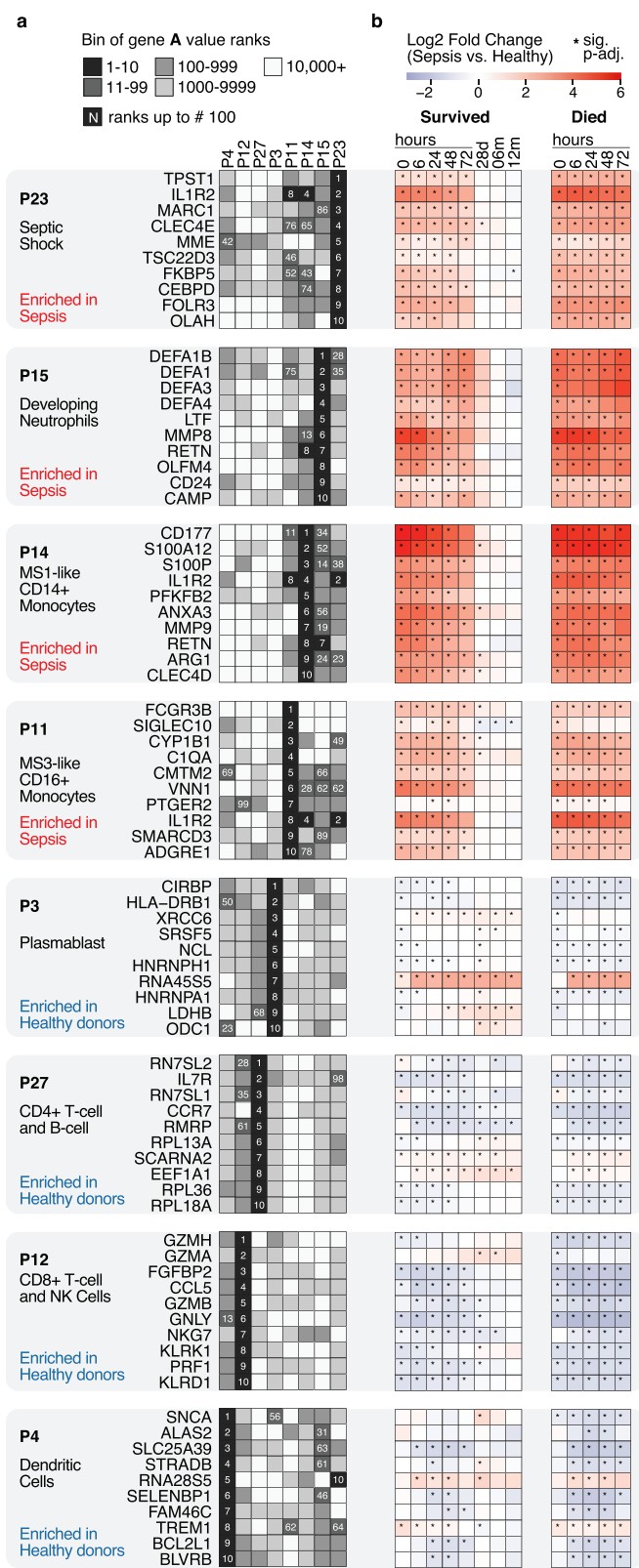

**Fig. 4 | Top gene amplitude (A) values for CoGAPS patterns enriched in sepsis subjects and healthy donors. a** Binned rank of top CoGAPS gene amplitude values (A-values) for patterns associated with sepsis (P23, P15, P14, P11) and those associated with healthy subjects (P3, P27, P12, P4). The intensity of the color indicates the binned rank of the gene per pattern. The number (white) indicates the rank of the gene within the pattern up to rank 100. **b** Gene expression changes (same genes as in (**a**)) comparing sepsis versus healthy) for survivors and non-survivors over time (times: 0, 6, 24, 48, 72 h, 28 days, 6 and 12 months). The star (*) marks a significant difference (≤0.05) in comparison where the p-value was determined using the Welch Two Sample t-test and adjusted for multiple comparisons using the Benjamini & Hochberg method. Please see Supplementary Data 2 for information about full results with completed statistical results.

proteins compared to all other CoGAPS patterns consistent with a role in MS1 induction (Fig. 2f). During our comparative analysis, we also found that prognostic CoGAPS pattern 15 is enriched for a class of granulocytes in severe COVID-19 that have been designated "developing neutrophils" that express genes characteristic of low density immature pre- and pro- neutrophils such as *DEFA3, CAMP* and *LCN2* (Fig. 5a–e)[25,28]. Low-density neutrophils can have immunosuppressive properties in severe infections[30]. The top genes that define CoGAPS patterns 14 (MS1 monocytes) and 15 (granulocytes\developing neutrophils) are shared and include *RETN, S100P, MMP8/9*, and *CD177* (Fig. 4a). These genes are consistent with those identified in previous reports that define dysfunctional monocytes and neutrophils in sepsis[31,32]. Closer inspection of the expressed genes specific to recently reported classes of sepsis-associated immunosuppressive neutrophil populations shows that they are regulated in our cohort and enriched in patterns 14 and 15 (Fig. 5d, e and Supplementary Fig. 7a, b)[32]. Taken together, the lymphopenia and neutrophilia detected by hematology and CIBERSORT along with the cell states revealed through latent space exploration are consistent with impaired cellular immunity in the Ghanaian cohort.

## CoGAPS patterns map disease trajectories

Current models of sepsis suggest concurrent roles for immunosuppressive and pro- and anti-inflammatory responses[33]. CoGAPS decomposition of the Ghanaian RNA sequencing data reveals these phenotypes as sepsis subjects transition through illness and recovery. Included are dynamic processes in the lymphoid and myeloid compartments as well as features linked to physiological states. Pattern 23 confers a highly elevated risk for poor outcomes while pattern 4 is protective (Supplementary Fig. 4b). These findings are consistent with the pattern 23 link to increased BUN, Sodium and Glucose and septic shock (Fig. 2e and Supplementary Fig. 3) versus the pattern 4 connection to the myeloid compartment in healthy donors (Supplementary Fig. 5b, e). Because Patterns 4 and 23 confer the greatest protection and risk we mapped these pattern values for every sample to inspect relationships (Fig. 6a). The healthy donors occupy an area defined by high values for pattern 4 and low pattern 23 values, and sepsis survivors progress towards this healthy field as they recover. In contrast, the average longitudinal trajectory of non-survivors shows little net movement and these subjects have increased pattern 23 risk and low pattern 4 protection values. Inspection of individual survivor trajectories reveals a subset that traverses the high-risk pattern 23 space during their disease course (Fig. 6b). In these subjects the pathogenic MS1 (P14) and developing neutrophil (P15) patterns decrease through time and the lymphocyte patterns 12 and 27 recover (Fig. 6c). Taken together these results support a clinical management strategy where both immunotherapeutic interventions that target cellular features defined by these CoGAPS patterns and supportive therapies to manage shock best described by pattern 23 would promote improved sepsis outcomes (Fig. 6d).

Fig. 6a, b). MS1 cells display a dysregulated response to antigenic stimuli and inspection of genes that characterize the MS1 state, including *RETN* and *IL1R2*, reveals that they are top contributors to pattern 14 and are regulated through time (Fig. 4a, b). MS1 cells can be induced from hematopoietic stem and progenitor cells by IL6 and IL10[27]. Pattern 14 has the greatest correlation with IL10 in our analysis of circulating

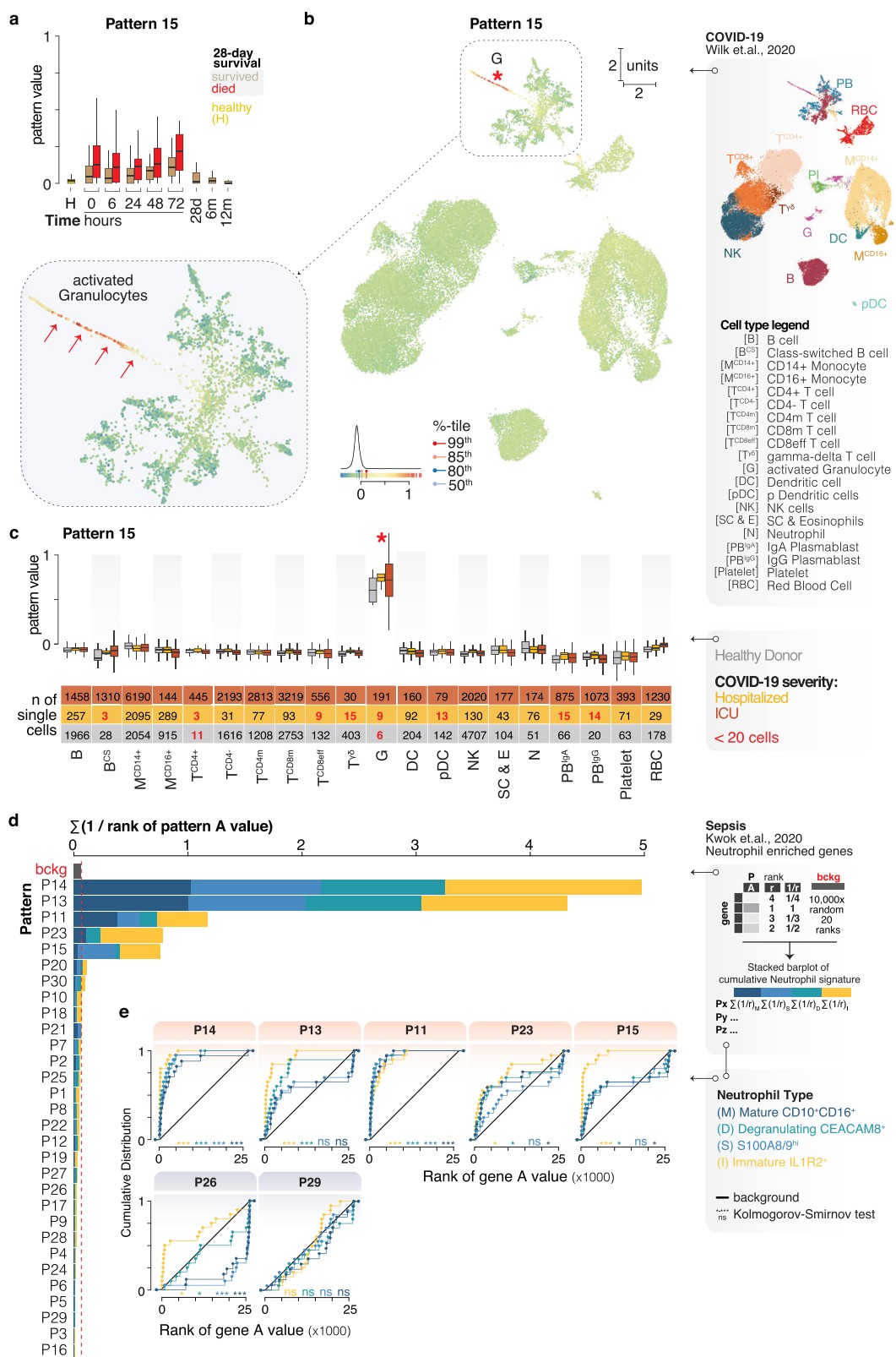

## Discussion

The human host is a sensitive instrument for detecting and monitoring infection. The extent to which a "diseased" host-response differs from normal healthy donors, and their degree of recovery over time correlates with clinical outcomes[34]. Sepsis subjects have been stratified by host-response using large multi-omics datasets and more recently, the development and application of single-cell RNA sequencing has

revealed novel host cell populations linked to sepsis and severe inflammatory disease[4,5,24,25]. Notably each of these studies implicate immunophenotypes as key features related to poor outcomes. However, to date, no rationally developed therapeutics targeting specific immunological mechanisms approved for sepsis are in wide use. In addition, data from LMICs and geographically diverse locations is severely limited and presents a critical gap towards the development

**Fig. 5 | Neutrophil-related signatures in the Ghana sepsis cohort. a** Boxplot of CoGAPS pattern 15 values in subjects who died by 28 days (red) and survivors (tan) through time. **b** Projection of single-cell RNA-Seq data describing immune cells in COVID-19 (Wilk et al., 2020, PMID 32514174) onto pattern 15 values. The color gradient bar−range of projected values, percentile pins (%-tile)−select percentile cutoffs, and the histogram−distribution of all pattern values. Enlargement highlights region of activated granulocytes (G)/developing neutrophil cells (red arrows). **c** Boxplot of pattern 15 values for different cell types. Color fill corresponding to healthy donors (gray, cells of 6 subjects), hospitalized COVID-19 subjects (orange, cells of 1 subject), or COVID-19 subjects in ICU (dark red, cells of 6 subjects). The table below the plot shows the exact number of cells in each group. The group with less than 20 cells is highlighted with red text. The asterisk (*) labels the most prominent cell type. **d** Stacked barplot showing the cumulative rank of enriched genes associated with four different neutrophil types identified in Kwok et al., 2023 (PMID 37095375, Fig. 2b, up in sepsis) in each of CoGAPS patterns described in this study. Neutrophil types: immature IL1R2+ (gold), degranulating CEACAM8+ (cyan), S100A8/9hi (light blue), and mature CD10 + CD16+ (dark blue). The top 20 sepsis-enriched genes in Kwok et al., were used to identify the neutrophil subtypes. The red dashed line indicates the background signal (bckg, see "Methods" for details). **e** Cumulative distribution plot of the neutrophil subtype enriched genes. Stars indicate significant differences from the background model calculated where ns = not significant, * ≤0.05, **≤ 0.01, *** ≤0.001. The p-values were calculated using the exact two-sample Kolmogorov–Smirnov test (two-sided). Sample numbers (n) and statistical values for figure panels are described in Supplementary Data 2. All boxplots describe the median (middle horizontal line), 1st and 3rd quartiles (bottom and top of box, respectively), and data minimum and maximum (vertical whiskers).

of generalizable solutions for effective sepsis clinical management strategies[35]. Here, using a combination of empirical and bioinformatic approaches we show that the host response in a West African Ghanaian cohort monitors immunophenotypes of significance including lymphopenia and the CD14 + MS1 monocyte and developing neutrophil populations detected in sepsis and severe COVID-19[27–29]. Using the CoGAPS matrix factorization tool we also defined physiologic gene expression phenotypes. CoGAPS patterns 4 and 23 best define this axis where pattern 4 describes the myeloid compartment in healthy donors and pattern 23 shock and organ failure. Inspection of transcripts that contribute to the CoGAPS gene expression phenotypes include those previously linked to immunosuppression and sepsis outcomes. Genes high in the CD8 + T-cell\NK-cell pattern 12 include GZMH (CTLA1), GZMK and TIGIT[31,36] and IL7R in the CD4 + T-cell pattern 27[8] (Fig. 4). Our data suggest that interventions that target treatable traits to promote patterns 4 and 27 would be associated with improved outcomes (Fig. 6d and Supplementary Fig. 4c). Notably, pattern 27 is high in T-cell cell signatures and there are multiple ongoing efforts to modulate T-cell activity and numbers including IL-7 immunoadjuvants and PD-L1 antibodies[37–39]. Examination of genes highly specific to individual patterns also supports translational opportunities for new interventions. Protective pattern 4 is uniquely characterized by the ALOX15 gene, a lipoxygenase that catalyzes the generation of specialized proresolvin mediators with anti-inflammatory properties that are being evaluated for therapeutic potential[40]. New insight into mechanisms of dysregulated myelopoiesis in sepsis also offers opportunities for treatable traits. We noted that our protein data from plasma is consistent with results showing that IL10 is linked to induction of the monocytic MS1 cell state and future immunomodulators could target this pathway to mitigate sepsis pathophysiology[27].

Cellular diversity and abundance are increasingly recognized as key drivers of patient subgroups in inflammatory disease. Recently reported cell-type abundance phenotypes (CTAPs) in rheumatoid arthritis defined by selectively enriched cell types are dynamic and both monitor and predict response to treatment[41]. Our longitudinal data reflect similar dynamic changes in gene expression phenotypes in sepsis subjects. Identifying complementary features of sepsis endotypes suitable for evaluation in clinical trials is an intense area of active investigation[9]. Although our results in the Ghanaian cohort reproduce and generalize shared immunophenotypes of significance to sepsis, we do not more broadly define endotypes in this West African cohort or if the CoGAPS phenotypes confer risk and protection across patient subtypes. The subjects described here do have an increased probability of being assigned to the reported SRS1 endotype at enrollment versus SRS2, but pathogen and population diversity have been shown to impact the performance of genomic stratification tools[42]. There is an extreme burden of multidrug-resistant bacteria and malaria in this hospitalized population in West Africa as well as emerging pathogens not often included in sepsis host response studies[43,44]. Future studies will assess if the Ghanaian

cohort reveals new endotypes characterized by distinct functional or pathophysiological mechanisms.

Immunosuppression occurs both in the early sepsis and in survivors with continued chronic illness. Progress toward effective clinical management strategies to improve outcomes will require approaches to map and interpret immunological states during sepsis progression and intervention throughout the entire course of disease and recovery. Improved time to care and emergency treatment can mitigate the immediate life-threatening complications of sepsis but those that survive often experience secondary infections and exhibit long-term cognitive and clinical sequelae with reduced 5-year survival[45]. Our analysis suggests that gene expression patterns in 28-day survivors are shared with healthy donors, but it is not clear if there are subjects in this West African cohort that continue to display molecular and cellular phenotypes consistent with prolonged immunosuppression. We note that individual subjects move between the reported SRS1 and SRS2 endotypes but a complete molecular dissection of immunological dynamics to map continued chronic human sepsis pathology will require more high-resolution longitudinal gene expression studies of the host-response (Supplementary Fig. 1e). To that end, we took advantage of the increasingly available number of datasets in the public domain to interpret the gene expression patterns defined by the CoGAPS patterns in the Ghana cohort. We have leveraged the NeMO analytics portal and gEAR platform[46] to visualize and explore our dataset alongside key public datasets. We anticipate that this interface will facilitate the integration of increasingly available information from next-generation platforms and drive new insights for therapeutic intervention by mapping phenotypes with treatable traits onto sepsis progression and recovery[47].

# Methods
## Ethics statement
Study protocol NMRC.2016.0004-GHA was approved by the Naval Medical Research Command (NMRC) Institutional Review Board in compliance with all applicable Federal regulations governing the protection of human subjects as well as host country IRBs. The protocol was approved by the Committee on Human Research, Publication, and Ethics (CHRPE) at Kwame Nkrumah University of Science & Technology. All procedures were in accordance with the ethical standards of the Helsinki Declaration of the World Medical Association. All patients, or their legally authorized representatives, provided written informed consent. Compensation was only provided to cover subject transportation costs for follow-up visits.

## Study cohorts
A detailed description of the sepsis cohort recruited at Komfo Anokye Teaching Hospital (KATH), in Kumasi, Ghana has been published in Blair et al., 2023[10]. The subset of subject demographic, clinical, and laboratory information relevant to this study are described in the Supplementary Data 1. The analysis was performed in the R[48]

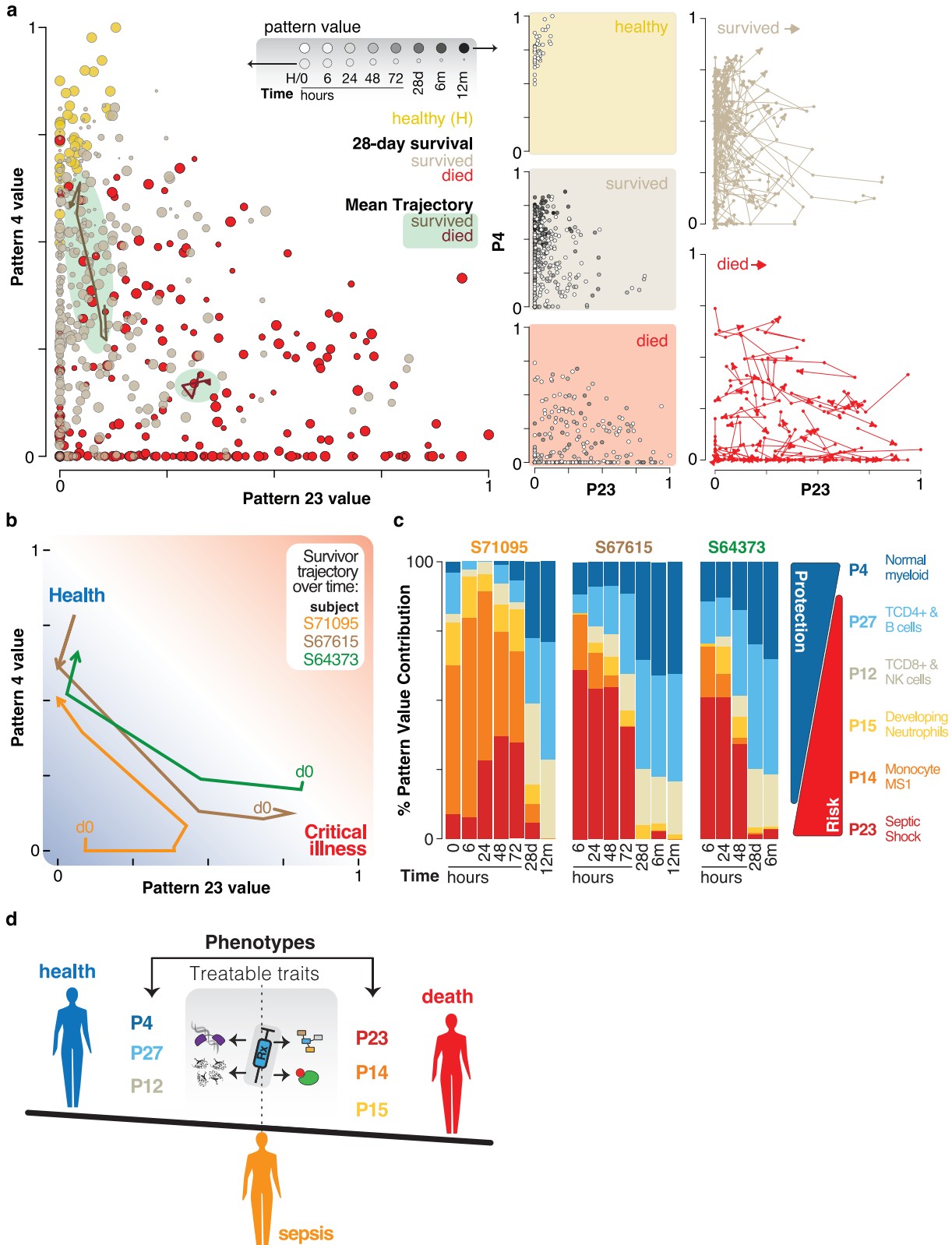

**Fig. 6 | CoGAPs gene expression immunophenotypes map sepsis trajectories.**
**a** Scatter plot comparing values of pattern 4 (associated with health) and pattern 23 (associated with septic shock). Point size indicates the collection times (0, 6, 24, 48, 72 h, 28 days, 6 and 12 months) and the point fill corresponds to healthy donors (H, gold), survivors (tan), and those who died by 28 days (red). The line and arrow with a green background correspond to the mean trajectory of the survivors (dark tan), and those who died by 28 days (dark red). The second column shows individual group data plots where the sample collection time is represented in greyscale. The last column shows the paths of all the survivors (tan), and non-survivors (red). **b** Selected surviving subject trajectories using pattern 23 and pattern 4. **c** Barplot showing the change in health-related patterns (P4, P27, P12) and sepsis-related patterns (P23, P14, P15) over time for select surviving subjects showing dynamic recovery over time. **d** Model highlighting the opportunities to target treatable traits of phenotypes revealed by analysis of gene expression patterns in this study that describe sepsis survival or death.

environment using the *table1*[49] package. Briefly, eligible subjects were identified at the admission department, the adult medical wards, or the emergency department. Inclusion criteria included subjects with suspected infection, meeting 2 or more systemic inflammatory response syndrome (SIRS) criteria, and those who were at least 18 years of age. Considering the parent cohort, subjects for this study were selected to make a balanced group considering reported sex, age, and 28-day survival. There was a total of 120 sepsis subjects (sepsis cohort) and 42 healthy donors (healthy cohort) whose samples were sequenced longitudinally at 0 (enrolment), 6, 24, 48, 72 h, 28 days, 6, and 12 months post enrolment. A complete set of samples for each subject and collection time point was not available due to loss to follow-up or death. A total of 573 samples from both cohorts were sequenced. Three of the 120 sepsis subjects did not have information about 28-day mortality and therefore were removed after CoGAPS pattern calculation.

### RNA sequencing

RNA sequencing was performed as reported in Rozo 2020[50]. RNA samples were quantified using the Qubit 2.0 Fluorometer (Life Technologies, Carlsbad, CA, USA), and RNA integrity was checked with 4200 TapeStation (Agilent Technologies, Palo Alto, CA, USA). rRNA depletion along with globin depletion was performed using Globin Zero Gold kit (Illumina, San Diego, CA, USA). RNA sequencing library preparation used NEBNext Ultra RNA Library Prep Kit for Illumina by following the manufacturer's recommendations (NEB, Ipswich, MA, USA). Briefly, enriched RNAs were fragmented for 15 minutes at 94 °C. First-strand and second-strand cDNA were subsequently synthesized. cDNA fragments were end-repaired and adenylated at 3'ends, and a universal adapter was ligated to cDNA fragments, followed by index addition and library enrichment with limited cycle PCR. Sequencing libraries were validated using the Agilent Tapestation 4200 (Agilent Technologies, Palo Alto, CA, USA), and quantified by using Qubit 2.0 Fluorometer (Invitrogen, Carlsbad, CA) as well as by quantitative PCR (Applied Biosystems, Carlsbad, CA, USA). The sequencing libraries were multiplexed and clustered on a flow cell and loaded on the Illumina HiSeq instrument according to the manufacturer's instructions. The samples were sequenced using a $2 \times 150$ Paired-End (PE) configuration to yield approximately 50 million, 150 bp, paired-end reads per sample. All time points for a given subject were processed and sequenced in the same batch except for 0h samples that were previously sequenced using the same protocol. Image analysis and base calling were conducted by the HiSeq Control Software (HCS). Raw sequence data (.bcl files) generated from Illumina HiSeq was converted into fastq files and de-multiplexed using Illumina's bcl2fastq 2.17 software. One mismatch was allowed for index sequence identification. Samples were aligned to the human genome build hg38 using STAR[51] and gene level information was quantified using RSEM[52].

### CoGAPS pattern identification and projections

The Bayesian non-negative matrix factorization (NMF) algorithm CoGAPS[20,21] was run on the entire bulk RNA sequencing data matrix to define 30 patterns. CoGAPS decomposed the data matrix of experimental observations, "D" into 2 matrices "P" and "A", hence D ~ P*A. Here, "D" is the log2 RNA sequencing TPM matrix, with genes as rows and samples as columns. "P" is the pattern matrix with patterns as rows and samples as columns in which each pattern contains values for each sample, i.e., the strength of that pattern in each sample. "A" is the amplitude matrix with genes as rows and patterns as columns, with values indicating the strength of involvement of a given gene in each pattern. In this way, the "A" values for the individual patterns provide a "recipe" for reconstructing the full pattern of gene expression for each gene. Principal component analysis of gene-level TPM from RNA sequencing data was done using the *prcomp()* function in R[48]. To define relationships between the datasets generated in this study and public datasets we used the *projectR* package[11,53]. This package identifies common relationships between a source dataset and a target dataset based on shared latent spaces such as gene lists generated from data decomposition techniques such as CoGAPS, principal component analysis, NMF and UMAP. The *projectR* package has been developed and validated to overcome technical and batch confounders in source and target datasets and to enable the user to quantify similarities between high-dimensional datasets. Example code utilizing *CoGAPS* and *projectR* can be found at our Github in the "Data availability" statement. Enrichment analyses were calculated via the *geneSetTest()* function in the *limma* Bioconductor package in R[54]. Projections into UMAP space were plotted using *ggplot2* (3.4.4)[55] and *ggrastr* (1.0.2)[56] R packages.

### Public data

Public datasets used for the comparison of molecular dynamics using projections were obtained from, Cazalis et al., 2014−GSE57065[13], Wilk et al., 2020−GSE150728[25], and Reyes et al., 2020[24] using the single-cell portal at the Broad Institute (SCP548). classify our subjects into sepsis response groups (SRS) initially described by Davenport et al., 2016[5], we used *SepstratifieR*[14] package in R. The obtained probability of group membership, probability values were then plotted using ggplot2 (3.4.4)[55] functionality in R. The genes defining the Adaptive, Coagulopathic, and Inflammopathic sepsis subtypes were obtained from the Supplemental data file−Table 6 in Sweeney et al., 2018[6]. The gene signature patterns corresponding to different MARS groups (1–4) were obtained from Supplementary Appendix−Sepsis endotype 140 gene classifier table in Scicluna et al., 2017[4]. Genes enriched in the different sub-populations of neutrophils were extracted from Fig. 2b heatmap in Kwok et al., 2023[32]. The top 20 sepsis-enriched genes were used to identify neutrophil subtypes. Only 17 genes for mature neutrophils were available in our dataset. The background model was generated by randomly pulling 20 ranks in 10,000 iterations for all 30 patterns. The mean background signature ranks were used for plotting. Source data for all public datasets is described in the Supplementary Data 2 and "Data availability" statement.

### Gene expression analysis

The three subjects with unknown 28-day mortality were removed from the analysis. The differential comparisons were performed for surviving subjects (Survived), and subjects who died by 28 days (Died) for each collection time point (0, 6, 24, 48, 72 hours [h], 28 days [d], 6 and 12 months [m]) versus healthy donors (N = 42). The Survived subject's numbers were: $N_{0h} = 60$, $N_{6h} = 58$, $N_{24h} = 52$, $N_{48h} = 42$, $N_{72h} = 35$, $N_{28d} = 41$, $N_{6m} = 25$, $N_{12m} = 14$. The Died subject's numbers were: $N_{0h} = 48$, $N_{6h} = 51$, $N_{24h} = 42$, $N_{48h} = 34$, $N_{72h} = 22$. See Supplementary Data 2 for additional information. To determine whether there is a significant difference between the groups, the *t.test()* base R function was used on raw TPM values to run the Welch Two Sample t-test and determine the p-value. To correct for multiple testing, the *p.adjust()* base R function was used with the Benjamini & Hochberg (BH) correction method. The data was then visualized in various plots generated using *ggplot2*[55] R package and finalized using Adobe Illustrator.

### Modeling

Scaling and centering of the subject pattern values (P) was performed using *scale()* function in base R (2022.02.2)[48]. The remainder of the analysis was performed in Python (3.9.12). Classification models were generated using *sklearn* python package (1.3.0). Odds ratios and confidence intervals were generated using *statsmodels*[57] python package (0.13.2). Data was stored using data frames using *pandas* (1.4.2) python package. Means and standard deviations were calculated using *numpy*[58] (1.21.5) python package. Visualizations were generated using *matplotlib* (3.5.1)[59] python package and finished using Adobe Illustrator.

Data was filtered to keep only the samples with information about 28-day mortality (N = 117 subjects). After scaling and centering, the data was split into training and testing sets with a 70/30 proportion. Repeated stratified k-fold cross-validation with 10 folds and 10 repeats was performed on the training set (70) consisting of all patterns, using a random forest classifier for 28-day mortality. This process was repeated 10 times, to vary the training and testing set populations. Variable importance was averaged over all runs and measured via Gini importance. The most important patterns, chosen based on a break in the variable importance data, were P23, P4, P27, P15, and P12.

Repeated stratified k-fold cross-validation with 10 folds and 10 repeats was performed on the testing set consisting of the most important patterns (30), using a random forest classifier for 28-day mortality. This process was repeated 10 times, to vary the training and testing set populations. Performance was averaged over all runs and measured via area under the ROC curve (AUROC). Performance was analyzed additively, starting with the most important pattern and adding additional patterns one by one. The variable importance and performance analysis was completed on two additional classifiers, logistic regression (LR) and support vector machine (SVM) with a linear kernel to ensure robust results. Variable importance for both classifiers was measured via model coefficients. The best performing set of patterns from random forest consisted of 4 patterns. Performance of the top 4 patterns from logistic regression and support vector machine classifiers was lower, so random forest remained the best classifier option. Logistic regression was run on the top patterns. 95% confidence intervals were generated on the resulting coefficients. The coefficients were exponentiated to generate odds ratios.

### Protein analysis
IL-7, IL-10, IL-16, IL-17A, IP-10, TNF-a, TRAIL, and IL-27 were assayed on MesoScale Discovery U-plex assay plates according to the manufacturer's protocols. Ang-2 and sPD-L1 were assayed on MesoScale Discovery R-plex assay plates according to the manufacturer's protocols. Assay plates were read using a MESO QuickPlex SQ120 reader and analyte levels were calculated using Discovery Workbench 4.0.12 software. Pentraxin-3 and sTREM-1 were assayed on Quantikine R&D Systems ELISA kits. Assay plates were read using a SpectraMax M3 and analyte levels were calculated using Softmax Pro 7 GxP.

### Statistics and reproducibility
The selection of the study subject is described in the "study cohort" methods section. No statistical method was used to predetermine the sample size. Samples for subjects that did not have 28-day mortality information were excluded from some comparisons. The information about the precise number of samples used in each analysis, results of different analyses, and related exact statistical values are described in Supplementary Data 2 and can be accessed at Zenodo (https://doi.org/10.5281/zenodo.10916993). In cases where multiple comparisons were performed, when appropriate, the p-value was used to calculate the adjusted p-value (p-adjust).

### Reporting summary
Further information on research design is available in the Nature Portfolio Reporting Summary linked to this article.

## Data availability
Raw sequence data and processed subject gene level data used in this study have been deidentified and deposited in dbGaP (Accession: phs003608.v1.p1) under restricted access in compliance with study informed consent and National Institutes of Health Human Subjects Protection guidelines. Data can be obtained following local IRB approval and with a letter of collaboration with the primary study investigator(s). The processed deidentified data used to perform analyses and generate figures throughout the study can be accessed in Zenodo (https://doi.org/10.5281/zenodo.10916993). A description of available processed and source data is provided in the Supplementary Information with this manuscript. Limited computational methods are available at GitHub (https://github.com/HJF-ACESO/Sepsis_Ghana/). Source data is provided, however, we have not provided the line-by-line scripts used for the generation and manipulation of the figure panels. These can be obtained by request to the corresponding author. Data visualization (https://nemoanalytics.org/p?l=ChenowethEtAl2024&g=CEBPA), and latent space exploration (https://nemoanalytics.org/p?p=p&l=ChenowethEtAl2024&c=GhanaSepsisCoGAPSp30&algo=nmf) can be accessed through the Neuroscience Multi-Omic Analytics (NEMO) Portal.

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

## Acknowledgements

This work is supported by the Defense Health Agency through the Joint West Africa Research Group and Combatting Antibiotic-Resistant Bacteria with programmatic oversight from the Military Infectious Diseases Research Program. This work was supported by Cooperative Agreement (N62645-14-2-0001) and work unit number 6000.RAD1.J.A0310. Data sharing and visualization via NeMO Analytics were supported by the Hearing Health Foundation (R.H.), NIDCD/NIH Intramural Research Program DC000094-01 (R.H.) and U01 DC19370 (R.H.). The views expressed in this article are those of the authors and do not necessarily reflect the official policy, or position of the Henry M. Jackson Foundation for the Advancement of Military Medicine Inc., Department of the Navy,

Department of Defense, nor the US Government. All patients or their legally authorized representatives provided written informed consent. KLS is an employee of the US government and AGL is a US NAVY Officer. This work was prepared as a part of their official duties. Title 17 U.S.C. §105 provides that 'Copyright protection under this title is not available for any work of the United States Government.' Title 17 U.S.C. §101 defines U.S. Government work as work prepared by a military service member or employee of the U.S. Government as part of that person's official duties.

## Author contributions

J.G.C. and C.C. conceived the study. Gene expression analysis and bioinformatics were performed by C.C., P.G., J.B., D.A.S., M.C., and L.C. Public dataset curation efforts were supported by C.C., P.G., A.C.K., and R.H. Feature selection and modeling was performed by E.C., R.M., M.F., and M.C. Data interpretation was performed by J.G.C., C.C., D.V.C., K.L.S., P.G., J.B., D.A.S., and S.K. Clinical interpretation contributed by P.W.B., A.F., G.O., and E.B. Clinical research implemented by A.E., A.F., A.G.L., A.O., I.B., D.A., A.A.A., G.O., J.G.C., and D.V.C. The manuscript was written by J.G.C., C.C., P.G., J.B., and D.A.S. with input from all authors.

## Competing interests

The authors declare no competing interests.
