## [Peer Review File · Nature Communications]

Gene expression signatures in blood from a West African sepsis cohort define host response phenotypesREVIEWER COMMENTS

Reviewer #2 (Remarks to the Author):

Chenoweth et al performed blood RNA-sequencing on a Ghanaian cohort of sepsis patients to characterize host immune signatures in individuals from low- and middle-income countries. They used a variety of computational methods, including principal components analysis, CIBERSORT deconvolution, and CoGAPS factorization, to identify gene patterns and clinical variables that associate with disease progression and mortality. The authors used transfer learning and analysis of external bulk and single cell RNA-seq datasets to validate and interpret the gene patterns from their primary analysis.

Overall, the manuscript is well written and their analysis is well executed. The multiple timepoints profiled and integration with external datasets are key strengths of their study. In addition, this manuscript provides a valuable resource to compare findings with existing studies of sepsis host responses, which were primarily conducted in Western countries.

Suggestions to improve the manuscript:

1. Septic patients often have long-term sequelae and sepsis survivors have poor overall prognosis. The authors mention in their discussion that it is not clear whether there were differences between 28-day survivors and healthy individuals. It appears in Fig 1A that there is some separation between transcriptomes from healthy individuals and sepsis survivors, primarily in PC2. Perhaps the authors could look into PC2 and perform a focused analysis of late timepoints from survivors vs. healthy to identify patterns that discriminate recovered patients from healthy individuals.
2. The study finds a couple of neutrophil-associated patterns in their analysis, but the single cell datasets utilized in their manuscript have limited representation of neutrophils. A scRNA-seq study of neutrophils from sepsis patients was published recently (Kwok et al, 2023). The reviewer suggests mapping these patterns on this dataset to aid their interpretation of their neutrophil patterns.
3. The authors mention in their introduction that a number of endotype classifications have been proposed for patients with sepsis. Do the authors see similar groupings in their datasets? Are there similarities between existing endotype genes and those found in their CoGAPS patterns?
4. The authors show that a combination of patterns P3, P4, P12, and P23 is predictive of 28d mortality in their cohort. Could the authors test if a composite gene signature derived from these patterns (perhaps by taking the top associated genes) performs similarly in other datasets? The authors could utilize additional datasets, such as those from a recent mortality meta-analysis of sepsis datasets (Sweeney, et al, 2018).

Minor comments:

1. The COVID-19 dataset analyzed is incorrectly attributed in the main and supplementary figures to Reyes et al. The data was generated in a study by Wilk et al.

Reviewer #3 (Remarks to the Author):

I am grateful for the opportunity to review this manuscript. The topic is of importance: there is a need to better understand the host response of sepsis and the authors are to be applauded that they have conducted a gene expression signature study in West Africa given the fact that most of these studies are performed in high-income settings. Cited references are adequate. However, this reviewer has a number of significant concerns.

Major

- The paper is difficult to follow. What is the main message/overall narrative of this paper? Is the goal to predict mortality or to define characteristics of subpopulations (eg. hyperinflammation vs. immune suppression) or to find a model that differentiates between viral and bacterial infection? Now, the authors have all incorporated this into one single paper, making it less concise and therefore less comprehensible. In line, there are several overly complicated statistical / mathematical methods used and graphs presented, which do not add to a robust narrative. Some of the graphs are just too difficult to fully understand with limited legends.
- The patients are not well characterized making any meaningful link to the clinical syndrome of sepsis very difficult. Although this reviewer appreciates that performing clinical research in some resource refrained settings can be challenging, just the basic clinical information is missing. Previous clinical studies from the teaching hospital in Kumasi have been great. A Table 1 with baseline characteristics is missing. What was the source and microbiological cause of sepsis of the included patients? At what department were the patients treated and for how long? What about comorbidities? Treatment etc. etc.

Other

- How were the 120 patients selected for RNAseq? Inclusion criteria are reported only very briefly; there is only a reference to another article which mentions 187 included patients from Ghana.
- Unclear what is meant by Supp Tab (line 70) and Supp File (line 93). There is one supplementary file with multiple supplementary figures, but no tables.
- For timepoints 0h and 6h the AUC of qSOFA alone is calculated and found to be better than the AUC of PC1. However, the AUC of qSOFA for 72h is not shown. What was this value and how does it compare to the AUC of PC1 at 72h?
- All the AUCs shown seem to be from a discovery/derivation cohort. It seems that they need to be validated as they could very well be overfitted?
- Have the authors considered survivorship bias in any of their analyses? It seems that PC1 at later timepoints predicts mortality better. How many non-survivors are left after 72 hours?
- There are no healthy controls displayed in Fig S2. How relevant is the trajectory of survivor's vs non-survivors if there are no healthy controls?
- With regards to Fig S3A: the second microarray batch overlap between bacterial and viral which makes one wonder if the PC1 really differentiates there?
- The following sentences could perhaps be better placed in results instead of the methods section (line 361 – 365): "Five of the 30 patterns were identified as the most informative in this model (P23, P4, P27, P12, 362 P3.) Pattern 23 and 4 were the top 2 most significant predictors of 28-day mortality in the random forest model which is consistent with their individual performance at both early and later timepoints using logistic regression. Both of these approaches random forest and traditional logistic regression support this result."
- With regard to the Legend of Fig. 2: it states "Would be good to add an indication in A of which 2 patterns are in B (* or arrows?)." I guess this just needs to be deleted?
- No abstract has been included in the version of this reviewer.
- The limited set of protein measurements: these are mentioned in the Methods but are mentioned only very briefly in the Results section of the main document.
- Some examples of excerpts that need more clarification. Line 91: Projection of public data from bacterial and viral infections indicates that PC1 describes cellular components of the immune response to bacterial infections (Fig S3A and B and Supp File 2). Line 158 – 160: These results indicate that dynamic gene expression patterns are linked to cellular components of the immune response while prognostic patterns that distinguish survivors and non-survivors at enrollment better capture physiology. Line 342-344: Projection of CoGAPS and principal component gene weights defines relationships between samples in a new data set given relationships discovered in the primary sepsis RNA-seq dataset.

Reviewer #4 (Remarks to the Author):

This work states that using dimension reduction and transfer learning techniques along with additional public RNA sequencing data, transcriptional phenotypes in a West-African cohort were

defined, which were shared across diverse cohorts and infection contexts. These gene expression phenotypes map a physiological landscape of host response to sepsis and support interventions that target immunophenotypes to promote improved sepsis outcomes.

I do not see significance to the field in this work because it does not provide an original method, it is the sum of a number of already available classical models in machine learning.

I do not see how does it compare to the established literature because there is no comparison to other related works.

I am afraid that there some flaws in the data analysis, interpretation and conclusions.

It is not completely clear to me whether the use of transfer learning in this work is actually the correct one.

It is widely understood that Transfer learning (TL) is a technique in machine learning in which knowledge learned from a task is re-used in order to boost performance on a related task. And the key word here is LEARNING. Some knowledge LEARNED in one domain is transferred to another specific task, which also involves LEARNING AND also FINE-TUNING, maybe.

For example, for image classification, knowledge gained while learning to recognize cats could be applied when trying to recognize lions. However, in this work there is not such a thing. There is no knowledge learnt and then transferred.

First of all, a simple PCA is done on all data.

Then, to assess the generality of the dynamic over the PC1, authors state that used "used transfer-learning techniques by feature mapping projection between two datasets. We downloaded public leukocyte gene expression data from healthy donors and subjects that progress to septic shock. Projection of these expression data onto PC1 of the Ghana cohort recapitulates the segregation of healthy controls from septic subjects". I really don't think that this can be called transfer learning. There is not learning at all! It is the projection of one dataset into another one, independent one.

Moreover, authors state that the total RNA from peripheral blood was sequenced from 120 subjects in Ghana.

However, there is no indication of sex and gender of these participants to the study. The authors have not complied with the following recommendations related to this point: it is expected that the title and/or abstract must indicate when findings apply to only one sex or gender. The methods section should include whether sex and/or gender were considered in the study design and whether sex and/or gender of participants was determined based on self-report or assigned (and methodology used). Data should be reported disaggregated for sex and gender where this information has been collected.

Then, to determine if PC1 is useful for predicting mortality, authors generated ROC curves using host gene expression PC1 values at different times. It is not specified with which model the ROC curves were built.

Authors used (CoGAPS) a sparse Bayesian non-negative matrix factorization (NMF) method that decomposes expression data into component patterns and can identify gene expression modules that are specific for sub-groups of patients, diseases, transient processes, etc.

Then, a standard Random Forest (RF) technique was used to determine important CoGAPS patterns. For this analysis data was split into training and test sets in the ratio of 70:30 respectively. This is another methodological flaw: several cross-validation folds, at least 3, should have been used.

I don't think there enough detail provided in the methods for the work to be reproduced, mainly because no source code data nor a repo url were provided.

To all reviewers: We appreciate the insightful suggestions and critical review of our work and are excited to provide a revised version of the manuscript titled "*Gene Expression Signatures in Blood that Predict Mortality in a West African Sepsis Cohort Define Host Response Phenotypes.*" Please find below the point-by-point responses to the reviewer's comments.

Reviewer #2 (Remarks to the Author):

Chenoweth et al performed blood RNA-sequencing on a Ghanaian cohort of sepsis patients to characterize host immune signatures in individuals from low- and middle-income countries. They used a variety of computational methods, including principal components analysis, CIBERSORT deconvolution, and CoGAPS factorization, to identify gene patterns and clinical variables that associate with disease progression and mortality. The authors used transfer learning and analysis of external bulk and single cell RNA-seq datasets to validate and interpret the gene patterns from their primary analysis.

Overall, the manuscript is well written and their analysis is well executed. The multiple timepoints profiled and integration with external datasets are key strengths of their study. In addition, this manuscript provides a valuable resource to compare findings with existing studies of sepsis host responses, which were primarily conducted in Western countries.

We thank the reviewer for their summary of our work and the recognition of the value of longitudinal sampling to define the host response to sepsis. Second, we are excited that the reviewer identified the knowledge gap from low-resource countries that is addressed in our study. This is a current area that key leaders in the field continue to identify as critical towards generalizing sepsis endotypes (van Amstel RBE, Kennedy JN, Scicluna BP, et al. *Uncovering heterogeneity in sepsis: a comparative analysis of subphenotypes. Intensive Care Med.* 2023;49(11):1360-1369).

Suggestions to improve the manuscript:

1. Septic patients often have long-term sequelae and sepsis survivors have poor overall prognosis. The authors mention in their discussion that it is not clear whether there were differences between 28-day survivors and healthy individuals. It appears in Fig 1A that there is some separation between transcriptomes from healthy individuals and sepsis survivors, primarily in PC2. Perhaps the authors could look into PC2 and perform a focused analysis of late timepoints from survivors vs. healthy to identify patterns that discriminate recovered patients from healthy individuals.

We agree that long-term sequelae are an important area of investigation when noting that there is significant mortality in survivors of sepsis in the 2-to-3-month period following initial presentation (Delano MJ, Ward PA. 2016 Jan;126(1):23-31. PMID: 26727230). Although we see this as outside the scope of the current study, our approaches presented here could be used to look for these signatures and we are actively pursuing this with larger sample sizes and a focus on long-term outcomes that require additional clinical adjudication. Specifically, regarding the observation in PC2, we have identified an RNA sequencing batch effect in this dimension that precludes analysis. However, the value of using the CoGAPS tool is that it effectively decomposes the data in a manner that accounts for any batch or technical effects (Stein-O'Brien GL, Clark BS, Sherman T, et al. 2021;12(2):203).

2. The study finds a couple of neutrophil-associated patterns in their analysis, but the single cell datasets utilized in their manuscript have limited representation of neutrophils. A scRNA-seq study of neutrophils from sepsis patients was published recently (Kwok et al, 2023). The reviewer suggests mapping these patterns on this dataset to aid their interpretation of their neutrophil patterns.

This is an excellent suggestion by the reviewer. In response we have assessed the contribution of the top differentially expressed features in four classes of neutrophils from the Kwok et al. manuscript in our CoGAPS gene expression patterns. We now show that the immature IL1R2 neutrophil features from Kwok et al. are enriched in our CoGAPS gene expression patterns that predict day 28 mortality, most notably Pattern 15. This is consistent with our transfer learning using datasets from Wilk AJ 2020 where they describe a cell type that they name “developing neutrophils” that is specifically enriched in CoGAPS pattern 15. These results have been added as Fig. 5d and Supplementary Fig. 7.

3. The authors mention in their introduction that a number of endotype classifications have been proposed for patients with sepsis. Do the authors see similar groupings in their datasets? Are there similarities between existing endotype genes and those found in their CoGAPS patterns?

This is an important question at the center of the paradigm shift that is aimed at moving beyond the “one size fits all” approach to improving sepsis outcomes. As the reviewer indicates, we highlight some of the important published studies in our introduction that have used genomic approaches to identify patient subgroups with shared mechanisms and clinical outcomes. A recent comparative analysis of these subphenotypes (van Amstel 2023) has recognized that a lack of data from low-income countries is a critical limitation to the field. Although a formal comparison of our results to these datasets is outside the scope of the study here, in this revision we did make significant new efforts to relate our data to the endotypes reported by Scicluna 2017, Davenport 2016, and Sweeney 2018.

To compare host gene expression in our Ghanaian cohort to the reported sepsis response signatures SRS1 and SRS2 from Davenport 2016, we used the published machine learning tool Sepstratifier (Cano-Gamez 2022). Using this algorithm, we showed that our cohort was most aligned with the high-mortality SRS1 group. Taking advantage of our longitudinal data we show that the probability of being in SRS1 decreases over time in subjects that survive in our cohort (Fig. 1g, h). Examination of subject-level data shows that some subjects move between SRS1 and SRS2 through time (Supplementary Fig. 1e).

Sweeney 2018 reports three endotypes - Adaptive, Coagulopathic, and Inflammopathic. Although we could not formally quantify membership in these groups for our data because the source gene-level data is not readily available, the fold-change versus healthy controls was available from Sweeney 2018. Therefore, we looked at how the fold-change versus healthy in our cohort at each timepoint compared to the reported fold-change versus sepsis in the Sweeney 2018 study. Qualitatively our cohort was aligned most closely to the Inflammopathic group, and we could see the differences between sepsis subjects and healthy controls resolve through time in our West-African cohort. These data have been added as Supplementary Fig. 1c.

Finally, we also assessed the MARS 1-4 groups from Scicluna 2017 in our dataset. We used gene expression change signatures (Discovery vs Healthy) published by the authors in the supplementary appendix table and correlated these values to the log₂ fold changes (Sepsis vs Healthy) of the same genes in the Ghanaian data set. At time 0h, our sepsis cohort subjects most resembled the MARS 4 group (Pearson $r=0.85$) followed by MARS 2, MARS3, and MARS1 (Pearson $r=0.45$). Over time we saw a similar pattern here as with other public data sets – with time and return to health MARS signatures became progressively diminished and, in some cases, even anticorrelated. All the results are shown in Supplementary Fig. 1.

4. The authors show that a combination of patterns P3, P4, P12, and P23 is predictive of 28d mortality in their cohort. Could the authors test if a composite gene signature derived from these patterns (perhaps by taking the top associated genes) performs similarly in other datasets? The authors could utilize additional datasets, such as those from a recent mortality meta-analysis of sepsis datasets (Sweeney, et al, 2018).

We recognize the interest and value of reducing the CoGAPS patterns into informative sets of genes or “biomarkers”, especially for translational applications. However, in this revision, we have reduced the emphasis on classifier development to highlight the focus on underlying biology and phenotypes. A formal feature selection and classifier development with external validation is outside of the scope of this work. However, we do think it is important and exciting to show that the data decomposition approaches used here can be translated. A statistic has been published called “patternMarkers” that is designed specifically to do this with CoGAPS patterns (Stein-O’Brien 2017). We used the top 10 genes from patternMarkers for CoGAPS Pattern 23 in a recursive feature elimination followed by an exhaustive feature selection, both using a logistic regression classifier with stratified 5-fold cross-validation selecting for the outcome of 28-day mortality. All time points were included. The gene TPST1 provided the best area under the ROC curve (AUROC) with a performance of 0.844. This provided comparable performance to the model in revised Supplementary Fig. 4c that uses patterns 15, 4, 27, and 23. This result with TPST1 is not included in this new revision because we have reduced the emphasis on prognostic modeling in this new version.

Minor comments:

1. The COVID-19 dataset analyzed is incorrectly attributed in the main and supplementary figures to Reyes et al. The data was generated in a study by Wilk et al.

We thank the reviewer for identifying this error. This has been corrected.

Reviewer #3 (Remarks to the Author):

I am grateful for the opportunity to review this manuscript. The topic is of importance: there is a need to better understand the host response of sepsis and the authors are to be applauded that they have conducted a gene expression signature study in West Africa given the fact that most of these studies are

performed in high-income settings. Cited references are adequate. However, this reviewer has a number of significant concerns.

We appreciate the interest of the reviewer in this study. Both reviewers 2 and 3 share our recognition of the important contribution of the host response to solving sepsis and note the gap in non-Western datasets that our work fills.

Major

- The paper is difficult to follow. What is the main message/overall narrative of this paper? Is the goal to predict mortality or to define characteristics of subpopulations (eg. hyperinflammation vs. immune suppression) or to find a model that differentiates between viral and bacterial infection? Now, the authors have all incorporated this into one single paper, making it less concise and therefore less comprehensible. In line, there are several overly complicated statistical / mathematical methods used and graphs presented, which do not add to a robust narrative. Some of the graphs are just too difficult to fully understand with limited legends.

We thank the reviewer for this high-level comment. In this new revision, we have made significant efforts to narrow the focus of the work to present a more robust narrative. Specifically, we address the major goals of the sepsis community to identify complementary phenotypes across diverse populations that are candidates for treatable traits or therapeutic intervention (Maslove 2022 and van Amstel 2023). Our work shows that through host-gene expression analysis, we can identify cellular immunophenotypes (patterns) of importance for sepsis outcomes in an understudied West African population. Furthermore, we have performed new analyses to relate our work to published sepsis subtypes to generalize these new findings. Finally, we have reduced the figure count and removed analyses that do not add significantly to the narrative, while adding graphics and improved legends to guide the reader through the study.

- The patients are not well characterized making any meaningful link to the clinical syndrome of sepsis very difficult. Although this reviewer appreciates that performing clinical research in some resource refrained settings can be challenging, just the basic clinical information is missing. Previous clinical studies from the teaching hospital in Kumasi have been great. A Table 1 with baseline characteristics is missing. What was the source and microbiological cause of sepsis of the included patients? At what department were the patients treated and for how long? What about comorbidities? Treatment etc. etc.

We thank the reviewer for recognizing the challenges of clinical research in diverse low-resource settings which impacts available clinical information. This Ghanaian cohort was previously described in Blair 2023 PMID 36806137 which highlights key characteristics of the population. We have now provided a Table 1 in the Supplementary Data 1 file which includes some basic characteristics of the subjects used in this study and added a Study Cohorts section to the methods. In addition, we have added references from our co-authors at KATH that provide insight into the Kumasi clinical population.

Other

- How were the 120 patients selected for RNAseq? Inclusion criteria are reported only very briefly; there is only a reference to another article which mentions 187 included patients from Ghana.

These 120 patients were selected out of the previously reported 187 using a nested case-control design based on 28-day mortality outcome considering age and sex. Other criteria included the availability of longitudinal biospecimens and RNA quality following specimen extraction as well as cost constraints for RNA sequencing.

- Unclear what is meant by Supp Tab (line 70) and Supp File (line 93). There is one supplementary file with multiple supplementary figures, but no tables.

We apologize for this confusion. This new revision has updated Supplementary Information and Data with appropriate identifiers.

- For timepoints 0h and 6h the AUC of qSOFA alone is calculated and found to be better than the AUC of PC1. However, the AUC of qSOFA for 72h is not shown. What was this value and how does it compare to the AUC of PC1 at 72h?

The Glasgow Coma Scale was not assessed at 72 hours in this Ghanaian cohort so unfortunately, we cannot calculate qSOFA and perform this comparison. These analyses have been removed from the new revision to better focus the manuscript on the phenotypes revealed by gene expression versus prognostic classifier development.

- All the AUCs shown seem to be from a discovery/derivation cohort. It seems that they need to be validated as they could very well be overfitted?

We have removed a significant portion of the modeling in this new revision to focus on the biological phenotypes versus classifier development as stated in the reply to the above comment.

- Have the authors considered survivorship bias in any of their analyses? It seems that PC1 at later timepoints predicts mortality better. How many non-survivors are left after 72 hours?

At 72 hours we have data available for 35 survivors and 22 non-survivors. We have removed all of the modeling using PCA weights as discussed in the previous two replies.

- There are no healthy controls displayed in Fig S2. How relevant is the trajectory of survivor's vs non-survivors if there are no healthy controls?

We have removed this analysis from the revised manuscript to simplify the narrative but have provided an explanation below. In Figure S2 from the initial submission, we were asking if the main variation described by PC1 in the Ghana cohort from the initial submission Figure 1 is conserved in another independent cohort from Cambodia (Blair 2023). To do this we used the transfer learning technique "projectR" that "exploits the fact that if two datasets share common latent spaces, a feature mapping between the two can identify and characterize relationships between the data defined by individual latent spaces." In the original version Figure 1 we show that PC1 in the Ghana gene expression dataset is a trajectory from disease to health and the inclusion of healthy controls is important to identify this transition. However, once this trajectory is learned or defined mathematically, we can ask how ANY

sample set maps onto this space, even without healthy controls using the “projectR” algorithm. We would hypothesize that the samples derived from Cambodian survivors at later time points would be separated from their earlier time points if the trajectory initially defined by the Ghana dataset was shared and generalizable. This was demonstrated in the original version of Figure S2 along the X-axis “Projected Ghana PC1.” To better clarify this use of “projectR” throughout the paper we have updated main Figure 1 with a graphical cartoon in Fig. 1c of the revised manuscript.

- With regards to Fig S3A: the second microarray batch overlap between bacterial and viral which makes one wonder if the PC1 really differentiates there?

To create a more robust narrative with a clear objective we have removed this analysis from the revised manuscript, however, we have provided the following in response to the reviewer’s question. Original manuscript Figure S3A is simply a PCA deconvolution of the public dataset we downloaded from GSE63990 Tsalik et al. 2016. As part of our quality assurance process, we look at the main variation in any public dataset before using it alongside our data. We noted here that the authors had a technical batch effect that accounted for the largest variation in the gene expression data. However, this batch effect seemingly did not preclude the authors in Tsalik et al. 2016 from using supervised analysis to generate bacterial and viral and non-infectious gene expression classifiers. For our purposes, because we used this public gene expression dataset in our transfer learning to interpret our CoGAPS gene expression patterns in the Ghana dataset, it was most transparent to treat these two batches independently as displayed in the original version Figure S3B and S7.

- The following sentences could perhaps be better placed in results instead of the methods section (line 361 – 365): “Five of the 30 patterns were identified as the most informative in this model (P23, P4, P27, P12, 362 P3.) Pattern 23 and 4 were the top 2 most significant predictors of 28-day mortality in the random forest model which is consistent with their individual performance at both early and later timepoints using logistic regression. Both of these approaches random forest and traditional logistic regression support this result.”

We have revised this modeling as per the suggestion of the last reviewer and added details to the methods but also included an updated legend in Supplementary Figure 4.

- With regard to the Legend of Fig. 2: it states “Would be good to add an indication in A of which 2 patterns are in B (* or arrows?).” I guess this just needs to be deleted?

We thank the reviewer for identifying this error. We have updated all the legends in the revised version of the manuscript.

- No abstract has been included in the version of this reviewer.

The abstract has now been provided.

- The limited set of protein measurements: these are mentioned in the Methods but are mentioned only very briefly in the Results section of the main document.

As noted by the reviewer this study included a limited analysis of circulating proteins which we used primarily to aid in the biological interpretation of the CoGAPS patterns. We have now added new content to the discussion to indicate how these protein results in combination with the gene expression data point to potential treatable traits or targets for sepsis.

- Some examples of excerpts that need more clarification. Line 91: Projection of public data from bacterial and viral infections indicates that PC1 describes cellular components of the immune response to bacterial infections (Fig S3A and B and Supp File 2).

We have removed this sentence as its content did not fit into the updated and streamlined version of the manuscript.

Line 158 – 160: These results indicate that dynamic gene expression patterns are linked to cellular components of the immune response while prognostic patterns that distinguish survivors and non-survivors at enrollment better capture physiology.

This sentence has been rewritten in a simplified version for clarity. *“These results indicate that CoGAPS gene expression patterns are linked to physiological and cellular components of the immune response in sepsis.”*

Line 342-344: Projection of CoGAPS and principal component gene weights defines relationships between samples in a new data set given relationships discovered in the primary sepsis RNA-seq dataset.

We have rewritten this sentence for clarity in the methods.

Reviewer #4 (Remarks to the Author):

We appreciate the reviewer’s critical reading and assessment of our manuscript. Below are responses to the evaluation and description of updates we have provided to better highlight the key contributions of our study to the sepsis field.

This work states that using dimension reduction and transfer learning techniques along with additional public RNA sequencing data, transcriptional phenotypes in a West-African cohort were defined, which were shared across diverse cohorts and infection contexts. These gene expression phenotypes map a physiological landscape of host response to sepsis and support interventions that target immunophenotypes to promote improved sepsis outcomes.

I do not see significance to the field in this work because it does not provide an original method, it is the sum of a number of already available classical models in machine learning.

We have updated the manuscript to better communicate the major contributions of this work to the sepsis field. The reviewer recognizes the approach in our study which is to apply a combination of proven bioinformatic approaches and methods to a unique high-dimensional dataset. The study design did not intend to develop an original analytic method, and this is better reflected in the revised version of the manuscript.

I do not see how does it compare to the established literature because there is no comparison to other related works.

We thank the reviewer for their suggestion to highlight comparisons to related works. Our goal with this work was to assess sepsis phenotypes in longitudinal datasets in a population that has been understudied. We have provided additional context, references, and data analysis to better communicate the advancement of this work for the sepsis field and in light of established literature. In this revision we made significant new efforts to relate our data to the sepsis endotypes reported by Scicluna 2017, Davenport 2016, and Sweeney 2018. Please see Fig. 1g, h and Supplementary Fig. 1c, d, e for these new analyses. Also, we compared our work to a recent publication by Kwok et al. 2023 that identifies sepsis-enriched neutrophil subtypes in Fig. 5d and Supplementary Fig. 7.

I am afraid that there are some flaws in the data analysis, interpretation and conclusions. It is not completely clear to me whether the use of transfer learning in this work is actually the correct one. It is widely understood that Transfer learning (TL) is a technique in machine learning in which knowledge learned from a task is re-used in order to boost performance on a related task. And the key word here is LEARNING. Some knowledge LEARNED in one domain is transferred to another specific task, which also involves LEARNING AND also FINE-TUNING, maybe. For example, for image classification, knowledge gained while learning to recognize cats could be applied when trying to recognize lions. However, in this work there is not such a thing. There is no knowledge learnt and then transferred.

As highlighted by the reviewer in a point above, our study uses a “sum” of previously developed tools to decompose and interpret host gene expression in sepsis subjects. One of these tools is the R package “projectR” which has been successfully applied to gene expression datasets in several peer-reviewed manuscripts to map target datasets into learned latent spaces (*Sharma G, et al. projectR: an R/Bioconductor package for transfer learning via PCA, NMF, correlation, and clustering. Bioinformatics. 2020;36(11):3592-3593.*). This package has been developed to leverage “the machine-learning subdomain of transfer learning that exploits the fact that if two datasets share common latent spaces, a feature mapping between the two can identify and characterize relationships between the data defined by individual latent spaces.” Here, one dataset is the source in which the latent space representation is learned, and another is the target that is mapped into the latent spaces learned in the source. This transfer learning by dimensionality reduction is also described by *Pan, S.J., Kwok, J.T., and Yang, Q. (2008). Transfer learning via dimensionality reduction. Proceedings of the Twenty-Third AAAI Conference on Artificial Intelligence. 677–682.* Our choice of this data analysis package was chosen so we could take advantage of the readily available, sophisticated, well-annotated, and defined public gene expression datasets in the sepsis and infectious disease field and learn from information relevant for our

understudied Ghanaian population cohort. Our choice of the “transfer-learning” terminology was based upon these peer-reviewed published articles.

First of all, a simple PCA is done on all data.

Then, to assess the generality of the dynamic over the PC1, authors state that used "used transfer-learning techniques by feature mapping projection between two datasets. We downloaded public leukocyte gene expression data from healthy donors and subjects that progress to septic shock. Projection of these expression data onto PC1 of the Ghana cohort recapitulates the segregation of healthy controls from septic subjects". I really don't think that this can be called transfer learning. There is not learning at all! It is the projection of one dataset into another one, independent one.

We appreciate the reviewer restating the method, and apologize that it wasn't written clearly enough to impart its real meaning. We would like to refer the reviewer to above mentioned Pan et.al., 2008 paper describing the use of our transfer learning methods. To clarify further, we do not project dataset 1 into dataset 2, we learn dimensions from our data (PCA or NMF) and then transfer these dimensions into the second dataset (public), and then explore any dynamics in the new dataset.

Moreover, authors state that the total RNA from peripheral blood was sequenced from 120 subjects in Ghana. However, there is no indication of sex and gender of these participants to the study. The authors have not complied with the following recommendations related to this point: it is expected that the title and/or abstract must indicate when findings apply to only one sex or gender. The methods section should include whether sex and/or gender were considered in the study design and whether sex and/or gender of participants was determined based on self-report or assigned (and methodology used). Data should be reported disaggregated for sex and gender where this information has been collected.

We thank the reviewer for highlighting the lack of study cohort information. To ameliorate this shortcoming we have included additional descriptions of this study (Figure 1a, b, c), a sex-specific breakdown of our cohort and mortality (Figure 1d) and generated a `Table 1` summarizing cohort demographics (Supplementary Data 1). The details regarding the subject enrolment criteria and subject metadata have been previously published (Blair et.al., 2023). However, we now also include the 'Study Cohorts' section in the materials and methods providing basic cohort information. Sex or Gender were not the subject of this study, however, were still considered to ensure they do not introduce bias into our analyses.

Then, to determine if PC1 is useful for predicting mortality, authors generated ROC curves using host gene expression PC1 values at different times. It was specified with which model the ROC curves were built.

This sentence and analysis were removed along with a significant portion of the modeling. Per the suggestion of this and other reviewers, the manuscript was revised to simplify and streamline the focus on biological phenotypes versus classifier development.

Authors used (CoGAPS) a sparse Bayesian non-negative matrix factorization (NMF) method that decomposes expression data into component patterns and can identify gene expression modules that are specific for sub-groups of patients, diseases, transient processes, etc.

Then, a standard Random Forest (RF) technique was used to determine important CoGAPS patterns. For this analysis data was split into training and test sets in the ratio of 70:30 respectively. This is another methodological flaw: several cross-validation folds, at least 3, should have been used.

We thank the reviewer for identifying an issue with the modeling. We have revisited the Random Forest model and added 10 cross-validation folds as well as repeating the 70:30 test\train split 10 times. This new analysis is provided in Supplementary Fig. 4a along with a detailed “Modeling” methods section.

I don't think there enough detail provided in the methods for the work to be reproduced, mainly because no source code data nor a repo url were provided.

We recognize the limited methods of our previous submission. To address it, we significantly expanded the methods section adding information on cohorts, public data sets and their analysis, gene expression analysis, and updated modeling section. During this study, we did not create any new software and used previously published programming packages, all of which are detailed in the updated methods of the manuscript. The software was used according to published manuals and any specific settings are mentioned in the manuscript. To further promote scientific data sharing, we created a GitHub (https://github.com/HJF-ACESO/Sepsis_Ghana/) repository where we describe the strategy for the data decomposition and projections, as well as the code used for modeling. All the cohort metadata and raw data will be deposited in dbGAP repository and accession links published with the publication of the manuscript.

REVIEWER COMMENTS

Reviewer #2 (Remarks to the Author):

In their revised manuscript, Chenoweth et al incorporated additional analysis to understand the relationship of their sepsis cohort to published sepsis endotypes and the overlap between their gene program patterns and neutrophil transcriptional states. The authors have sufficiently addressed my concerns and I have no further comments.

Reviewer #3 (Remarks to the Author):

Thank you for the opportunity to review this revised manuscript. This reviewer recognizes and appreciates that the authors have simplified their narrative, resulting in a more concise article. However, this reviewer still has two remarks on the transfer learning approach, patient selection and the lack of clinical characteristics of the patients.

1. Re the transfer learning approach: Line 92: In order to assess generalizability of the dynamic PC1 pattern that is found, transfer learning is used to compare the data to publically available datasets. This reviewer would need some more background whether transfer learning is a valid technique to compare these datasets. In order to use transfer learning, the information learnt in the source model must be applicable to the target data. The authors have not provided any additional information on the methods used for this, for example whether any fine tuning was necessary and how this was done. Since the use of transfer learning has limited added value to this paper, this reviewer would politely suggest to consider to either omit the section or provide additional information on the methods used. If the use of transfer learning is not valid, the question arises of whether the CoGAPS patterns found in the Ghana cohort are applicable outside of this cohort. No formal validation was done without the use of transfer learning.
2. Re clinical data: The reviewer has not been able to find a Table 1 with clinical information pertaining to this transcriptome cohort. In PMID 36806137 (Blair et al.) clinical information is given of a larger cohort. It is key to include clinical information of the studied transcriptome cohort (n=120).
3. Other: Line 340: it remains unclear how the cohort was selected from parent cohort. It seems that patients were selected based on sex, age and survival. How was this done and why was this method chosen? Random sampling of patients might result in a less biased selection of patients.

Reviewer #4 (Remarks to the Author):

I do not see significance to the field in this work because it is just a simple pipeline for data processing. It does not provide an original method, it is the sum of a number of already available classical (also old) tools in machine learning. Authors themselves stated that "The study design did not intend to develop an original analytic method".

Moreover, the methodology and the choice of such tools is not justified at all. For example, why random forest and not neural networks, that are known to be better models? Or at least several classifiers should have been used and then compared, in order to have more robust results. Also PCA (30` s) for dimensional reduction, why? there are many other new and better methods (t-SNE, UMAP, ICA, etc).

The use of the term transfer learning is incorrect. Just because authors used an R package that has transfer learning in its name, it does not mean that what they have done is transfer learning. They have just applied a package for data projection.

To all reviewers: We appreciate the second round of suggestions following review of our work and are excited to provide a revised version of the manuscript titled “*Gene Expression Signatures in Blood that Predict Mortality in a West African Sepsis Cohort Define Host Response Phenotypes.*” Please find below the point-by-point responses to the reviewer's comments.

Reviewer #2 (Remarks to the Author):

In their revised manuscript, Chenoweth et al incorporated additional analysis to understand the relationship of their sepsis cohort to published sepsis endotypes and the overlap between their gene program patterns and neutrophil transcriptional states. The authors have sufficiently addressed my concerns and I have no further comments.

We thank the reviewer for prior constructive feedback and are glad we were able to satisfy all the requests.

Reviewer #3 (Remarks to the Author):

Thank you for the opportunity to review this revised manuscript. This reviewer recognizes and appreciates that the authors have simplified their narrative, resulting in a more concise article. However, this reviewer still has two remarks on the transfer learning approach, patient selection and the lack of clinical characteristics of the patients.

1. Re the transfer learning approach: Line 92: “*In order to assess generalizability of the dynamic PC1 pattern that is found, transfer learning is used to compare the data to publically available datasets*”. This reviewer would need some more background whether transfer learning is a valid technique to compare these datasets. In order to use transfer learning, the information learnt in the source model must be applicable to the target data. The authors have not provided any additional information on the methods used for this, for example whether any fine tuning was necessary and how this was done. **Since the use of transfer learning has limited added value to this paper, this reviewer would politely suggest to consider to either omit the section or provide additional information on the methods used.** If the use of transfer learning is not valid, the question arises of whether the CoGAPS patterns found in the Ghana cohort are applicable outside of this cohort. No formal validation was done without the use of transfer learning.

We would like to thank the reviewer for additional feedback. In response we have provided additional information on the rationale and methods used here for the transfer learning performed using the data analysis package “projectR” both in the main text and in the methods.

In layman’s terms, the innovation of “projectR” is the use of a mathematical mapping function defined from the latent spaces (or gene expression patterns) in a source data set, which enables the transfer of associated cellular phenotypes, annotations, and other metadata to samples in a target dataset. We would refer the reviewer to Stein-O’Brien 2019 PMID 31121116 where the sensitivity and specificity of “projectR” has formally been demonstrated and validated and the mathematical basis explained. All the packages including “projectR” were used with standard settings and no

tuning as communicated in the computational methods and code demonstration in our Github whose link is provided in the Data Availability Statement.

However, although the rationale and mathematics behind “projectR” are published in many peer-reviewed studies (Sharma 2020 PMID 32167521, Davis-Marcisak 2021 PMID 34376232), we do want to highlight to the reviewer that we do not rely solely on this transfer learning package for key conclusions and to show that the CoGAPS patterns identified in the source Ghana dataset are relevant to outside target datasets. To emphasize this point we have revised Figure 2 to highlight the complementary findings between computational approaches and clinical data. Indeed the “projectR” package was used to support the identity and interpretation of key cellular phenotypes, but the same conclusions can be seen in Figure 2 using hematology and CIBERSORT that use direct cell measures or completely different algorithms respectively.

Despite this, we do want to highlight the added value of the “projectR” transfer learning data analysis package to justify inclusion in our study. Although some of our conclusions could be independent of this approach as highlighted in Figure 2, this tool enabled us to readily look for cellular phenotypes that are not routinely measured with clinical hematology or accessible due to limits of analytical tools in low resource settings. In response to the reviewer’s concerns that we have no formal validation of these conclusions outside “projectR”, we have revised main Figure 5 to include a panel from the supplement where we used a different approach to classify neutrophil-associated CoGAPS patterns that validates and reinforces the transfer learning approach.

2. Re clinical data: The reviewer has not been able to find a Table 1 with clinical information pertaining to this transcriptome cohort. In PMID 36806137 (Blair et al.) clinical information is given of a larger cohort. It is key to include clinical information of the studied transcriptome cohort (n=120).

We would like to draw reviewers' attention to the file “Supplementary Data 1” that includes two tables that describe the transcriptome cohort split by day 28 mortality in tab 1 and gender in tab 2. Per the reviewer's suggestion, we have included additional clinical information to better describe the transcriptome cohort including demographics, medical history, clinical laboratory measures and sepsis clinical tool scores.

3. Other: Line 340: it remains unclear how the cohort was selected from parent cohort. It seems that patients were selected based on sex, age and survival. How was this done and why was this method chosen? Random sampling of patients might result in a less biased selection of patients.

We apologize to the reviewer for not providing sufficient information to clarify this selection process in our previous responses. We used a nested case-cohort design that is a well-defined method for selecting a subpopulation of a cohort study population (Prentice RL. A case-cohort design for epidemiologic cohort studies and disease prevention trials. *Biometrika*. 1986;73:1–11). A nested design is typically used when it is unnecessary or unfeasible to test a full cohort study population to answer a research question. The two primary nested study designs are nested case-

control and nested case-cohort. In a nested case-cohort design, a random selection of the cohort is selected plus all of the participants with the specific outcome of interest (for this research question the outcome of interest was 28-day mortality). We chose the nested case-cohort design because the random sub-cohort could be used in the future in combination with all of the participants with a different outcome of interest to constitute a new nested case-cohort study.

Reviewer #4 (Remarks to the Author):

I do not see significance to the field in this work because it is just a simple pipeline for data processing. It does not provide an original method, it is the sum of a number of already available classical (also old) tools in machine learning. Authors themselves stated that "The study design did not intend to develop an original analytic method".

We appreciate the reviewers' feedback. The objective of this study was to address a gap in the sepsis field that has been highlighted by key leaders. There is a critical lack of data from diverse populations and low to middle income countries with respect to sepsis phenotypes (van Amstel 2023 PMID 37851064, Mount 2023 PMID 37409899, Giamarellos-Bourboulis 2024 PMID 38168953). We use well-established tools (CoGAPS, NMF, projectR) to analyze data from an understudied population to fill this gap in the sepsis field.

Moreover, the **methodology** and the choice of such tools is not justified at all. For example, why random forest and not neural networks, that are known to be better models? Or at least several classifiers should have been used and then compared, in order to have more robust results. Also PCA (30` s) for dimensional reduction, why? there are many other new and better methods (t-SNE, UMAP, ICA, etc).

To address reviewer's request for comparison of other tools, we performed added analysis and provide a new Supplementary Figure 4d comparing the SVM: support vector machine, LR: logistic regression, and RF: random forest. When evaluating the performance of gene expression patterns to predict 28-day mortality the random forest feature selection revealed the best performance.

We thank the reviewer for their comment. PCA is used in a very limited way in this study as a first-pass tool to understand data complexity. In fact the main tool we employ throughout the manuscript relies on a nonnegative matrix factorization (NMF) tool CoGAPS and we specifically highlight the limitations of PCA on line 126 "PCA is limited by the constraint that components be orthogonal, and variability is maximized for the earliest components. This often leads to conflation where multiple biological effects can be contained in a single PC. Other tools are needed for the analysis of dynamic, high-dimensional data in heterogeneous illnesses such as sepsis to gain biological insight."

The use of the term transfer learning is incorrect. Just because authors used an R package that has transfer learning in its name, it does not mean that what they have done is transfer learning. They have just applied a package for data projection.

We have limited the designation of the approach as “transfer learning” throughout the manuscript.

We would like to highlight to the reviewer that the package “projectR” does not have transfer learning in the name, rather it has been deployed in multiple peer reviewed transfer learning papers where identified latent spaces in a source dataset have been used to interrogate target datasets with validation (Sharma 2020 PMID 32167521, Stein-O’Brien 2019 PMID 31121116, Davis-Marcisak 2021 PMID 34376232).

REVIEWERS' COMMENTS

Reviewer #4 (Remarks to the Author):

OK with the responses.